# Preference-Calibrated Optimization with Score-Level Distribution Alignment for Text-to-Image Diffusion Model Unlearning

Xiuyuan Wang [1]   Weiming Liu [1]   Hongyu Cai [1]   Xin Gao [2]   Fan Wang [1]   Chaochao Chen [1]   Xiaolin Zheng [1]

## Abstract

While text-to-image diffusion models achieve remarkable generation quality, they inadvertently memorize sensitive content, necessitating machine unlearning to prevent undesired outputs. However, existing unlearning methods rely on suboptimal surrogate objectives rather than directly optimizing the unlearning goal, leading to a fundamental objective mismatch. Moreover, these methods preserve model utility via surface-level constraints on model parameters or outputs, yet fail to capture the intrinsic generative dynamics of diffusion models, consequently triggering catastrophic forgetting. To address these challenges, we propose Preference-calibrated Optimization with Score-level Distribution Alignment (**POSDA**), a unified unlearning framework that harmonizes effective erasure with fine-grained structural preservation. Specifically, we reframe unlearning as a preference optimization problem by constructing a reward that explicitly quantifies the unlearning objective. Additionally, we introduce score-level distribution alignment to ensure the invariance of the underlying manifold topology of the unlearned model, thereby preventing distributional drift. Extensive experiments across object, style, and NSFW unlearning tasks demonstrate that POSDA achieves state-of-the-art erasure efficacy while maintaining superior model utility compared to existing methods.

## 1. Introduction

Recent advancements in text-to-image diffusion models (DMs) have enabled the generation of diverse and realistic images conditioned on textual prompts. However, since these models are trained on large-scale, uncurated datasets (e.g., LAION (Schuhmann et al., 2022) and COYO (Byeon et al., 2022)), they inevitably learn to generate sensitive content, including Not Safe for Work (NSFW) content, copyrighted artworks, and personal identities, posing significant ethical and legal concerns.

To mitigate the potential negative societal impact, *machine unlearning* has emerged as a critical solution (Li et al., 2025a;b; Wang et al., 2025c). The goal of unlearning is to selectively remove a model's ability to generate certain concepts while preserving its generative capabilities across the remaining concepts. Despite the growing body of research in this area, existing methods face two key challenges that hinder effective unlearning.

**First**, existing methods primarily rely on heuristic *surrogate objectives* that treat unlearning as a supervised mapping task, rather than optimizing the unlearning goal directly. For instance, some methods (Gandikota et al., 2023; Zhang et al., 2024a) force the model to map the forget concept's noise predictions to those of a "neutral" concept, while others (Orgad et al., 2023; Arad et al., 2024; Chavhan et al., 2025; Biswas et al., 2025) edit model parameters to inhibit specific activations. However, given the high-dimensional parameter space and complex generative mechanisms of diffusion models, the validity of these handcrafted surrogate objectives cannot be guaranteed. Therefore, these approaches create a fundamental **objective mismatch** between the optimization target and the true unlearning goal. **Second**, to preserve the utility of the unlearned model, prior works typically enforce constraints on parameter distance (Kumari et al., 2023; Sendera et al., 2025), image similarity (Wu et al., 2024; Huang et al., 2024), or noise prediction errors (Gandikota et al., 2024; Lu et al., 2024). However, these *surface-level constraints* fail to preserve the intrinsic structure of diffusion models as they ignore the underlying *generative dynamics*. Consequently, existing methods are insufficient to prevent distributional drift during unlearning and ultimately result in **catastrophic forgetting** on retained concepts.

To address these challenges, we propose **POSDA** (**P**reference-calibrated **O**ptimization with **S**core-level **D**istribution **A**lignment), a unified unlearning framework that harmo-

---

[1]Zhejiang University, Hangzhou, China [2]Tongji University, Shanghai, China. Correspondence to: Xiaolin Zheng <xlzheng@zju.edu.cn>.

*Proceedings of the 43rd International Conference on Machine Learning*, Seoul, South Korea. PMLR 306, 2026. Copyright 2026 by the author(s).

nizes effective erasure with fine-grained structural preservation. First, to address objective mismatch, we reframe unlearning as a **preference optimization** problem based on the reinforcement learning (RL) paradigm. Specifically, we formulate the iterative denoising process as a Markov Decision Process (MDP), where the diffusion model functions as a policy. Instead of relying on surrogate objectives, we directly optimize the policy using a dynamic anchor-based reward that explicitly quantifies the unlearning goal. Second, to prevent catastrophic forgetting, we propose **score-level distribution alignment**. Recognizing that diffusion models fundamentally approximate score functions (Song & Ermon, 2019), we directly minimize the score-level divergence between the unlearned and original models on the retain set. This ensures that the intrinsic generative dynamics and manifold topology remain invariant for non-target concepts.

We summarize our main contributions as follows:

- We propose POSDA, a novel framework that reformulates unlearning as a preference optimization problem. We incorporate a dynamic anchor-based reward that explicitly measures the efficacy of unlearning to directly steer the diffusion policy.
- We introduce score-level distribution alignment to preserve the intrinsic generative dynamics of the diffusion process, preventing catastrophic forgetting.
- Extensive experiments on object, style, and NSFW unlearning tasks demonstrate that POSDA achieves state-of-the-art performance.

## 2. Related Work

Existing unlearning methods for text-to-image diffusion models, situated within broader efforts on robust representation learning and recommendation (Wang et al., 2024; 2025b;a; 2026; Liu et al., 2025b;a), can be classified into four mainstreams:

**Data Curation and Inference-Time Methods.** Early methods focus on dataset filtering during pre-training, such as removing harmful content (Stability AI, 2022) or utilizing licensed datasets (Rao, 2023). Alternatively, inference-time guardrails intercept malicious input via text classifiers (Yang et al., 2024a), or block inappropriate outputs using safety checkers (Rombach et al., 2022). Other inference-time strategies (Schramowski et al., 2023; Brack et al., 2023) inject negative guidance to suppress concepts during sampling. However, retraining is computationally prohibitive, while inference-time methods leave the latent knowledge intact, rendering them vulnerable to attacks (Yang et al., 2024b; Tsai et al., 2024).

**Finetuning-Based Methods.** To address these limitations, research has shifted toward model finetuning for permanent concept erasure. Pioneering studies (Gandikota et al., 2023;

Kumari et al., 2023; Wu et al., 2024) maximize the denoising error on forget concepts to steer the model away from unwanted behaviors. Subsequent works enhance unlearning efficacy by incorporating auxiliary constraints, such as saliency masks (Fan et al., 2023), statistical shifting (Alberti et al., 2025), anchor optimization (Bui et al., 2025), or task-specific pairwise data (Park et al., 2024). To mitigate the general utility degradation caused by global parameter updates, other methods focus on localized modifications, such as editing text embeddings (Arad et al., 2024), redirecting cross-attention (Zhang et al., 2024a), or inserting lightweight adapters (Lyu et al., 2024). Concurrently, knowledge distillation frameworks (Zhou et al., 2025; Huang et al., 2024; Chen et al., 2025; Heng & Soh, 2023) have been adopted to balance forgetting and retention objectives.

**Parameter Editing-Based Methods.** Parameter editing-based unlearning directly modifies weights via non-iterative updates. Some methods (Orgad et al., 2023; Gandikota et al., 2024; Lu et al., 2024) utilize closed-form solutions to remap cross-attention layers, while others leverage spectral analysis (Lee et al., 2025) or neuron pruning (Chavhan et al., 2025; Sendera et al., 2025; Cywiński & Deja, 2025) to nullify critical weight components responsible for target concepts.

**Adversarial Training for Robust Unlearning.** To address the vulnerability of unlearning to concept recovery attacks, adversarial training frameworks (Zhang et al., 2024b; Kim et al., 2024; Srivatsan et al., 2025; Gao et al., 2025) integrate attack simulations into the optimization loop. By iteratively generating and suppressing adversarial prompts, these methods enhance unlearning robustness against jailbreaks.

Despite these advancements, existing methods primarily rely on heuristic surrogate objectives, rather than directly optimizing the unlearning goal, and therefore fail to guarantee thorough and robust elimination of target concepts. In contrast, our method directly aligns the model with the unlearning goal via preference optimization, ensuring robust erasure while preserving intrinsic generative dynamics through score-level distribution alignment.

## 3. Method

In this section, we propose the POSDA framework, which harmonizes effective concept erasure with fine-grained structural preservation. Following the preliminaries, we explicitly formulate unlearning as a preference optimization problem guided by dynamic anchors. Then, to prevent catastrophic forgetting, we explicitly introduce a score-level distribution alignment mechanism that maintains the intrinsic generative dynamics of the original model. Finally, we integrate these components into a unified optimization framework within a reinforcement learning paradigm.

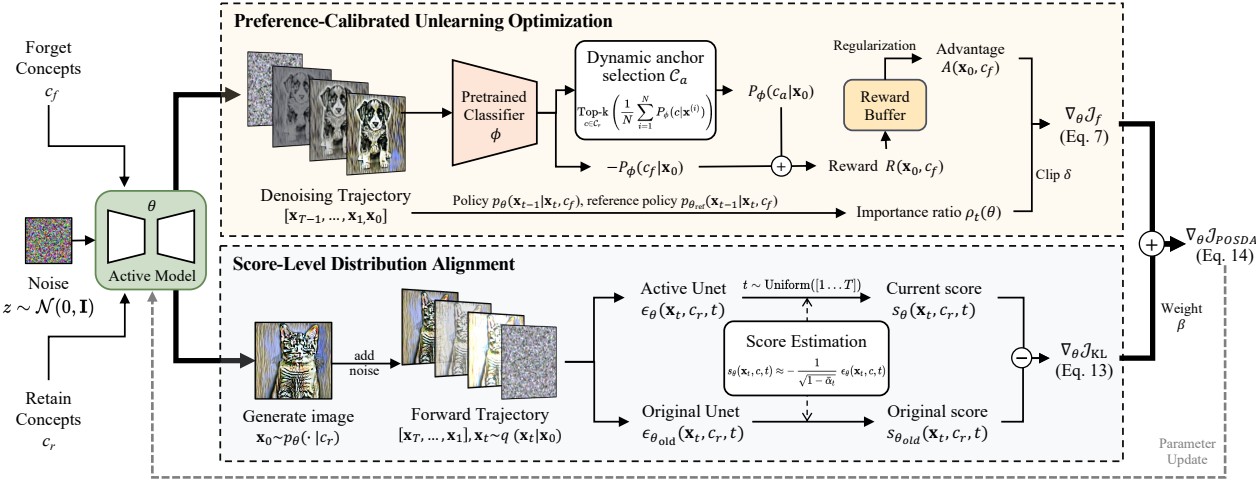

*Figure 1.* **Overview of POSDA framework.** (Top) Preference-Calibrated Optimization: We formulate unlearning as an MDP where a dynamic anchor-based reward guides the diffusion policy to suppress $c_f$ while aligning with the nearest semantic neighbors $\mathcal{C}_a$. (Bottom) Score-Level Distribution Alignment: To prevent catastrophic forgetting, we align the score estimates of $\theta$ with frozen $\theta_{old}$ on $c_r$. The final objective aggregates these gradients to harmonize targeted erasure with structural preservation.

### 3.1. Preliminaries

**Latent Diffusion Models.** In this work, we focus on Stable Diffusion (Rombach et al., 2022), a prominent Latent Diffusion Models (LDMs), which learns a conditional distribution $p_\theta(\mathbf{x}_0|c)$ over images $\mathbf{x}_0$ given a context $c$ (e.g., text prompts). Unlike standard diffusion models operating in pixel space, LDMs perform the diffusion process in a compressed latent space. During the forward process, the latent variable $\mathbf{x}_0$ encoded from real data is gradually perturbed through: $q(\mathbf{x}_t|\mathbf{x}_{t-1}) = \mathcal{N}(\mathbf{x}_t; \sqrt{\alpha_t}\mathbf{x}_{t-1}, (1-\alpha_t)\mathbf{I})$, where $\alpha_t$ is a predefined noise schedule. A neural network $\epsilon_\theta(\mathbf{x}_t, c, t)$ is trained to predict the noise added during the forward process by minimizing the following objective:

$$\mathbb{E}_{t\sim\text{Uniform}([1...T]),\epsilon\sim\mathcal{N}(0,\mathbf{I}),\mathbf{x}_t\sim q(\mathbf{x}_t|\mathbf{x}_{t-1})}[\|\epsilon - \epsilon_\theta(\mathbf{x}_t, c, t)\|^2].$$

During inference, the model samples an initial latent $\mathbf{x}_T \sim \mathcal{N}(0, \mathbf{I})$ and iteratively applies the reverse process $p_\theta(\mathbf{x}_{t-1}|\mathbf{x}_t, c)$ until obtaining $\mathbf{x}_0$, producing a reverse denoising trajectory $\{\mathbf{x}_T, \mathbf{x}_{T-1}, \ldots, \mathbf{x}_0\}$. Each reverse step is parameterized as an isotropic Gaussian:

$$p_\theta(\mathbf{x}_{t-1}|\mathbf{x}_t, c) = \mathcal{N}(\mathbf{x}_{t-1}|\mu_\theta(\mathbf{x}_t, c, t), \sigma_t^2\mathbf{I}), \quad (1)$$

where $\mu_\theta(\mathbf{x}_t, c, t) = \frac{1}{\sqrt{\alpha_t}}(\mathbf{x}_t - \frac{1-\alpha_t}{\sqrt{1-\bar{\alpha}_t}}\epsilon_\theta(\mathbf{x}_t, c, t))$, and $\bar{\alpha}_t = \prod_{i=1}^t \alpha_i$.

**Problem Definition: Concept Unlearning.** Consider a pre-trained LDM parameterized by $\theta_{old}$ as the original model for unlearning. We define two disjoint sets of concepts: a forget set $\mathcal{C}_f$ containing concepts to be erased (e.g., specific objects, styles, or NSFW content), and a retain set $\mathcal{C}_r$

containing concepts to be preserved (e.g., general or neighboring concepts). During training and evaluation, these concepts are embedded into text prompts using standard templates (detailed in Appendix B). The goal of unlearning is to learn a model $\theta$ that minimizes the generation ability of concepts $c_f \in \mathcal{C}_f$, while ensuring the distribution $p_\theta(\cdot|c_r)$ remains close to $p_{\theta_{old}}(\cdot|c_r)$ for $c_r \in \mathcal{C}_r$.

### 3.2. Dynamic Anchor-based Preference Modeling for Unlearning

**Formalizing the Unlearning Objective.** Since unlearning aims to prevent the model from generating forget concept $c_f$, a natural approach is to minimize the semantic alignment between the forget concept and the corresponding generated images. However, commonly used general text-image alignment metrics, such as CLIP scores (Radford et al., 2021) and PickScore (Kirstain et al., 2023), often lack the granularity required to distinguish fine-grained concepts. As shown in Figure 2, we visualize generated samples for the target style ("Abstractionism") using the original (Col. 1) and unlearned models (Col. 2), alongside reference images from other styles (Cols. 3-4). We observe that images from distinct styles receive highly similar CLIP score and PickScore with respect to the same prompt, indicating that these generic metrics lack sufficient discriminative ability for fine-grained concept unlearning.

To address this ambiguity, we employ a discriminative classifier $\phi$ trained specifically to distinguish the target and neighboring concepts. Let $P_\phi(y|\mathbf{x})$ denote the posterior probability of a generated image $\mathbf{x}$ belonging to a specific concept label $y$. A naive strategy is to define the unlearn-

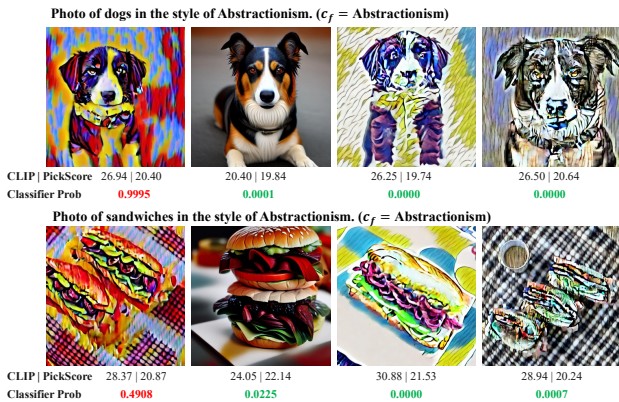

*Figure 2.* **Comparison of potential reward signals.** Generic text-image alignment metrics fail to distinguish fine-grained styles, whereas a task-specific classifier provides highly discriminative signals for unlearning.

ing reward as $R_{\text{naive}}(\mathbf{x}, c_f) = -P_\phi(c_f|\mathbf{x})$, which steers the generated samples away from the decision boundary of $c_f$. However, this reward suppresses the target concept without specifying a valid alternative semantic attractor, thereby causing excessive distribution shifts.

**Dynamic Anchor Selection.** To counterbalance the unbounded repulsion, it is necessary to provide an "anchor" to guide the distribution shift. Previous works (Gandikota et al., 2023; 2024; Lu et al., 2024) often rely on manually designed static anchors, such as mapping the forget concept to a generic "neutral" concept or an empty string. However, these heuristically defined anchors neglect complex semantic features and intrinsic model distribution, inevitably forcing the optimization along a high-energy trajectory that distorts the data manifold and degrades generation quality. Therefore, we propose a robust and **dynamic anchor selection mechanism**, which adaptively identifies the nearest semantic neighbor for each forget concept, thereby minimizing the distribution displacement required for unlearning. Specifically, given a forget concept $c_f$, we first generate a set of images $\mathcal{X}_{c_f} = \{\mathbf{x}^{(1)}, \mathbf{x}^{(2)} \dots, \mathbf{x}^{(N)}\}$ with model $\theta$ using diverse prompts associated with $c_f$. We then identify the anchor set $\mathcal{C}_a$ by selecting the top-$k$ retain concepts with the highest posterior probabilities:

$$\mathcal{C}_a = \text{Top-k}_{c\in\mathcal{C}_r}\left(\frac{1}{N}\sum_{i=1}^{N}P_\phi(c|\mathbf{x}^{(i)})\right). \quad (2)$$

**Final Reward Design.** Leveraging the dynamic anchor set $\mathcal{C}_a$, the final reward function $R(\mathbf{x}, c_f)$ is designed to suppress the forget concept while maximizing the alignment with anchor concepts, as follows:

$$R(\mathbf{x}, c_f) = \sum_{c\in\mathcal{C}_a} P_\phi(c|\mathbf{x}) - P_\phi(c_f|\mathbf{x}). \quad (3)$$

By maximizing Eq. (3), we impose a "soft" constraint that steers the generation away from the forbidden region while towards the intrinsic data manifold anchored by $\mathcal{C}_a$.

### 3.3. Preference-Calibrated Unlearning Optimization

To utilize the proposed unlearning reward for model optimization, we formulate the denoising process of a conditional diffusion model as a finite-horizon MDP with horizon $T$ (Black et al., 2024). At each timestep $t$, the state is defined as $s_t = (c, t, \mathbf{x}_t)$, the action corresponds to the previous latent $a_t = \mathbf{x}_{t-1}$, and the policy is given by the reverse transition $\pi_\theta(a_t|s_t) = p_\theta(\mathbf{x}_{t-1}|\mathbf{x}_t, c)$. A trajectory $\mathbf{x}_{T:0}$ thus corresponds to a complete denoising path conditioned on prompt $c$.

**Reinforcement Objective for Unlearning.** Given $\mathcal{C}_f$, we optimize the diffusion policy to minimize the generation of forget concepts by maximizing the expected unlearning reward:

$$\mathcal{J}_f(\theta) = \mathbb{E}_{c_f\sim\mathcal{C}_f,\mathbf{x}_0\sim p_\theta(\mathbf{x}_0|c_f)}\big[R(\mathbf{x}_0, c_f)\big] \quad (4)$$

However, the reward is only available at the end of the image generation process, making it non-trivial to identify which intermediate denoising decisions contribute positively to the final unlearning objective. To address this, we propagate the terminal reward backward along the denoising trajectory by optimizing the expected log-likelihood of each reverse diffusion step weighted by the final reward, as follows:

$$\nabla_\theta\mathcal{J}_f(\theta) = \mathbb{E}\Big[R(\mathbf{x}_0, c_f)\sum_{t=1}^{T}\nabla_\theta\log p_\theta(\mathbf{x}_{t-1}|\mathbf{x}_t, c_f)\Big], \quad (5)$$

where the expectation is taken over $c_f \sim \mathcal{C}_f$ and trajectories $\mathbf{x}_{0:T} \sim p_\theta(\cdot|c_f)$. Detailed proof is provided in Appendix A.

**Prompt-wise Advantage Normalization.** To reduce reward scale variability across different prompts and to obtain a more stable advantage signal, we maintain a prompt-specific reward history buffer $\mathcal{B}_c = \{R_c^{(1)}, \dots, R_c^{(n)}\}$ for each conditioning prompt $c$. After each generation step, the terminal reward $R(\mathbf{x}_0, c)$ is appended to the corresponding buffer $\mathcal{B}_c$, which is updated in a first-in-first-out manner to retain recent reward statistics. Based on this buffer, we normalize the raw reward into a standardized advantage estimate $A(\mathbf{x}_0, c) = (R(\mathbf{x}_0, c) - \mu_c)/\sigma_c$, where $\mu_c = \mathbb{E}_{r\sim\mathcal{B}_c}[R]$ and $\sigma_c = \sqrt{\text{Var}_{r\sim\mathcal{B}_c}[R]}$ are the moving mean and standard deviation of the buffer, respectively. This normalization ensures that the optimization focuses on the relative quality of the generation rather than the absolute reward scale.

**Importance Weighted Policy Gradient.** To improve the stability of policy optimization, we rewrite the policy gradient objective using importance sampling with respect to the reference policy $p_{\theta_{\text{ref}}}$, which is periodically synchronized to

the current policy.

$$\nabla_\theta \mathcal{J}_f(\theta) = \mathbb{E}_{p_\theta(\mathbf{x}_{0:T}|c_f)} \left[ \sum_{t=1}^{T} A(\mathbf{x}_0, c_f) \nabla_\theta \log p_\theta(\mathbf{x}_{t-1}|\mathbf{x}_t, c_f) \right]$$

$$\approx \mathbb{E}_{p_{\theta_{\text{ref}}}(\mathbf{x}_{0:T}|c_f)} \left[ \sum_{t=1}^{T} A(\mathbf{x}_0, c_f) \rho_t(\theta) \nabla_\theta \log p_\theta(\mathbf{x}_{t-1}|\mathbf{x}_t, c_f) \right]$$

$$= \mathbb{E}_{p_{\theta_{\text{ref}}}(\mathbf{x}_{0:T}|c_f)} \left[ \sum_{t=1}^{T} A(\mathbf{x}_0, c_f) \nabla_\theta \rho_t(\theta) \right], \quad (6)$$

where $\rho_t(\theta) = \frac{p_\theta(\mathbf{x}_{t-1}|\mathbf{x}_t, c_f)}{p_{\theta_{\text{ref}}}(\mathbf{x}_{t-1}|\mathbf{x}_t, c_f)}$ denotes the importance ratio at timestep $t$. To further prevent excessively large policy updates, inspired by PPO (Schulman et al., 2017), we impose a clipping constraint on the importance ratio. The final unlearning objective is defined as:

$$\nabla_\theta \mathcal{J}_f(\theta) = \mathbb{E} \left[ \sum_{t=1}^{T} A(\mathbf{x}_0, c_f) \nabla_\theta \text{clip} \left( \rho_t(\theta), 1 - \delta, 1 + \delta \right) \right]. \quad (7)$$

## 3.4. Score-Level Distribution Alignment for Structure Preservation

Preference optimization effectively suppresses the forget concept, yet it risks disrupting the model's behavior on retain concepts. As illustrated in Figure 3 (b), while the unconstrained optimization successfully vacates the target distribution (evidenced by the disappearance of the red region), it concurrently causes a significant distributional shift for the retain concepts. To mitigate this, we propose **Score-Level Distribution Alignment**, which explicitly constrains the unlearned model $\theta$ to maintain the generative dynamics of the frozen original model $\theta_{\text{old}}$ on the retain set.

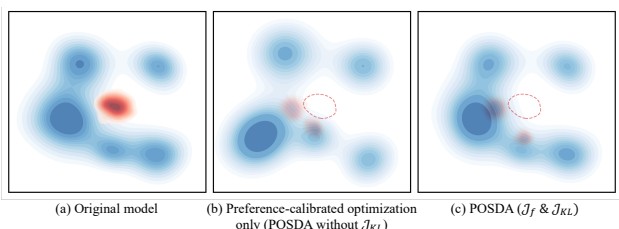

| (a) Original model | (b) Preference-calibrated optimization only (POSDA without $\mathcal{J}_{KL}$) | (c) POSDA ($\mathcal{J}_f$ & $\mathcal{J}_{KL}$) |

*Figure 3.* **Sample distributions for forget (Red) and retain (Blue) concepts.** Unconstrained optimization severely shifts retain distributions, whereas our proposed score-level alignment anchors distributions to the original manifold, harmonizing erasure with structural preservation.

**Objective Formulation.** We formulate the alignment objective as minimizing the Kullback-Leibler (KL) divergence between the generated distributions of the unlearned model and the original model over $\mathcal{C}_r$.

$$\mathcal{J}_{KL}(\theta) = \mathbb{E}_{\mathbf{x}_0 \sim p_\theta(\cdot|c_r)} \left[ \log \frac{p_\theta(\mathbf{x}_0|c_r)}{p_{\theta_{\text{old}}}(\mathbf{x}_0|c_r)} \right]. \quad (8)$$

Directly optimizing the KL divergence in the clean data space is analytically intractable and numerically unsta-

ble. Since the distributions may exhibit limited or non-overlapping support after unlearning, their densities become ill-defined, leading to unreliable gradients. To address this, we adopt a score-based approximate replacement by optimizing over noisy marginals across diffusion timesteps (Wang et al., 2023). The injected Gaussian noise acts as a smoothing mechanism that ensures distributional overlap and yields well-behaved gradients. Let $p_{\theta,t}(\mathbf{x}_t|c_r)$ denote the marginal distribution of the perturbed sample $\mathbf{x}_t$ at timestep $t$. Consequently, we derive a stable and tractable surrogate objective:

$$\mathbb{E}_{t \sim \text{Uniform}([1...T]), \mathbf{x}_0 \sim p_\theta(\cdot|c_r)} \left[ \log \frac{p_{\theta,t}(\mathbf{x}_t|c_r)}{p_{\theta_{\text{old}},t}(\mathbf{x}_t|c_r)} \right], \quad (9)$$

where $\mathbf{x}_t$ is sampled via fixed forward perturbation $q(\mathbf{x}_t|\mathbf{x}_0) = \mathcal{N}(\mathbf{x}_t; \sqrt{\bar{\alpha}_t}\mathbf{x}_0, (1-\bar{\alpha}_t)\mathbf{I})$ on the generated sample $\mathbf{x}_0$. Note that $\mathbf{x}_0$ is generated by the unlearned model $\theta$ via a deterministic sampling process to ensure differentiability, allowing gradients to propagate back to $\theta$.

**Gradient Estimation via Score Matching.** However, optimizing Equation (9) remains challenging because, unlike the transition probability $p_\theta(\mathbf{x}_{t-1}|\mathbf{x}_t)$ that admits a closed-form Gaussian expression, the marginal term $p_{\theta,t}(\mathbf{x}_t|c_r)$ represents an implicit density at timestep $t$ whose gradient corresponds to the score function. Consequently, we compute its gradient with respect to $\theta$ as:

$$\mathbb{E} \left[ \left( \nabla_{\mathbf{x}_t} \log p_{\theta,t}(\mathbf{x}_t|c_r) - \nabla_{\mathbf{x}_t} \log p_{\theta_{\text{old}},t}(\mathbf{x}_t|c_r) \right) \frac{\partial \mathbf{x}_t}{\partial \theta} \right]. \quad (10)$$

**Proposition 3.1.** *Given a latent diffusion model with conditioning context $c$, the score function of the noisy latent variable $\mathbf{x}_t$ satisfies*

$$s_\theta(\mathbf{x}_t, c, t) \triangleq \nabla_{\mathbf{x}_t} \log p_{\theta,t}(\mathbf{x}_t|c) \approx -\frac{1}{\sqrt{1-\bar{\alpha}_t}} \epsilon_\theta(\mathbf{x}_t, c, t).$$

*Proof.* By definition, for a Gaussian forward diffusion process, the score function with respect to $\mathbf{x}_t$ is

$$\nabla_{\mathbf{x}_t} \log q(\mathbf{x}_t \mid \mathbf{x}_0) = -\frac{1}{1-\bar{\alpha}_t} (\mathbf{x}_t - \sqrt{\bar{\alpha}_t}\mathbf{x}_0). \quad (11)$$

Using the standard reparameterization $\mathbf{x}_t = \sqrt{\bar{\alpha}_t}\mathbf{x}_0 + \sqrt{1-\bar{\alpha}_t}\,\epsilon$, where $\epsilon \sim \mathcal{N}(0, \mathbf{I})$, we can derive $\mathbf{x}_t - \sqrt{\bar{\alpha}_t}\mathbf{x}_0 = \sqrt{1-\bar{\alpha}_t}\,\epsilon$. Substituting this into Equation (11) yields $\nabla_{\mathbf{x}_t} \log q(\mathbf{x}_t \mid \mathbf{x}_0) = -\frac{1}{\sqrt{1-\bar{\alpha}_t}}\,\epsilon$.

Since the model $\epsilon_\theta(\mathbf{x}_t, c, t)$ is trained to predict $\epsilon$ via minimizing $\mathbb{E}_{\mathbf{x}_0, \epsilon, t} \left[ \|\epsilon - \epsilon_\theta(\mathbf{x}_t, c, t)\|^2 \right]$, the score function can be approximated by

$$s_\theta(\mathbf{x}_t, c, t) \approx -\frac{1}{\sqrt{1-\bar{\alpha}_t}} \epsilon_\theta(\mathbf{x}_t, c, t), \quad (12)$$

as required. $\qquad\square$

Based on Proposition 3.1 and chain rule, computing the gradient of the intermediate perturbed state $x_t$ with respect to $\theta$ requires backpropagating through the generated sample $x_0$. Since $x_0$ is obtained via a multistep reverse diffusion process, which is a composition of deterministic denoising steps, we formalize this full sampling trajectory as a differentiable mapping $x_0 = f_\theta(x_T, c_r) = \left( \mu_\theta^{(1)} \circ \mu_\theta^{(2)} \circ \cdots \circ \mu_\theta^{(T)} \right)(x_T, c_r)$, where $\mu_\theta^{(t)}$ denotes the deterministic denoising update at timestep $t$. Accordingly, the gradient $\frac{\partial x_0}{\partial \theta} = \frac{\partial f_\theta}{\partial \theta}$ is computed by automatic differentiation through the unrolled sampler. By applying the chain rule $\frac{\partial x_t}{\partial \theta} = \frac{\partial x_t}{\partial x_0} \frac{\partial x_0}{\partial \theta} = \sqrt{\bar{\alpha}_t} \frac{\partial f_\theta}{\partial \theta}$, we can calculate Equation (10) as:

$$\nabla_\theta \mathcal{J}_{KL}(\theta) = \mathbb{E}\Big[ \sqrt{\bar{\alpha}_t}(s_\theta(\mathbf{x}_t, c_r, t) - s_{\theta_{\text{old}}}(\mathbf{x}_t, c_r, t)) \frac{\partial f_\theta}{\partial \theta} \Big]$$
$$\approx \mathbb{E}\Big[ \frac{\sqrt{\bar{\alpha}_t}}{\sqrt{1 - \bar{\alpha}_t}} (\epsilon_{\theta_{\text{old}}}(\mathbf{x}_t, c_r, t) - \epsilon_\theta(\mathbf{x}_t, c_r, t)) \frac{\partial f_\theta}{\partial \theta} \Big]. \quad (13)$$

Consequently, Equation (13) acts as a regularization force that pulls the generated $\mathbf{x}_0$ in a direction that minimizes the discrepancy between the noise predictions of the unlearned model and the original model, thereby preserving the original generative structure.

### 3.5. Overall Optimization

Finally, we integrate the dynamic anchor-based preference-calibrated optimization and the score-level distribution alignment into a unified framework, POSDA. The final objective function can be written as:

$$\nabla_\theta \mathcal{J}_{\text{POSDA}}(\theta) = \nabla_\theta \mathcal{J}_f(\theta) - \beta \nabla_\theta \mathcal{J}_{KL}(\theta), \quad (14)$$

where $\nabla_\theta \mathcal{J}_f(\theta)$ is the gradient of the preference-calibrated unlearning objective maximizing anchor-based reward (detailed in Equation (7)), $\nabla_\theta \mathcal{J}_{KL}(\theta)$ is the gradient of the KL divergence objective minimizing distribution divergence (detailed in Equation (13)), and hyperparameter $\beta > 0$ controls the strength of the distribution alignment regularization. The complete training procedure is summarized in Algorithm 1.

## 4. Experiments

In this section, we present a comprehensive evaluation of our proposed framework, POSDA, benchmarking its performance against state-of-the-art methods across three distinct unlearning scenarios: object, style, and NSFW concept erasure. Furthermore, we conduct ablation studies to analyze the individual contributions of our key components. Additional qualitative results are provided in Appendix C.

### 4.1. Experiment Setting

**Tasks and Datasets.** To comprehensively evaluate the effectiveness of our proposed framework, we conduct experiments to unlearn various concepts from the pre-trained

---

**Algorithm 1** POSDA Unlearning Procedure

1: **Input:** Original diffusion model $\theta_{\text{old}}$, classifier $\phi$, forget set $\mathcal{C}_f$, retain set $\mathcal{C}_r$, hyperparameters $k, \delta, \beta, \lambda$.
2: **Initialize:** $\theta \leftarrow \theta_{\text{old}}$, reward buffers $\mathcal{B}_c$ for $c \in \mathcal{C}_f$.
3: **while** train **do**
4:     **if** start of epoch **then** $\theta_{\text{ref}} \leftarrow \theta$
5:     Sample batch $c_f$ from $\mathcal{C}_f$, $c_r$ from $\mathcal{C}_r$.
6:     ▷ **Preference-Calibrated Unlearning Optimization**
7:     Sample trajectory $\mathbf{x}_{0:T} \sim p_{\theta_{\text{ref}}}(\mathbf{x}_{0:T}|c_f)$.
8:     Select dynamic anchors $\mathcal{C}_a$ using Equation (2).
9:     Compute reward $R(\mathbf{x}_0, c_f)$ using Equation (3).
10:     Save reward to $\mathcal{B}_c$ and compute advantage $A(\mathbf{x}_0, c_f)$.
11:     Compute unlearning gradient $\nabla_\theta \mathcal{J}_f(\theta)$ using Equation (7)
12:     ▷ **Score-Level Distribution Alignment**
13:     Sample images $\mathbf{x}_0 \sim p_\theta(\cdot|c_r)$.
14:     Perform forward perturbation $\mathbf{x}_{0:T} \sim q(\mathbf{x}_{0:T}|\mathbf{x}_0)$.
15:     Compute KL gradient $\nabla_\theta \mathcal{J}_{KL}(\theta)$ using Equation (13).
16:     ▷ **Update Model Parameters**
17:     Compute final gradients $\nabla_\theta \mathcal{J}_{\text{POSDA}}$ using Equation (14).
18:     $\theta \leftarrow \theta + \lambda \cdot \nabla_\theta \mathcal{J}_{\text{POSDA}}(\theta)$
19: **end while**
20: **Output:** Unlearned diffusion model $\theta$.

---

Stable Diffusion models, including object-related concepts, artistic styles, and NSFW attributes. Following existing studies (Zhang et al., 2024c; Cywiński & Deja, 2025), we utilize two widely adopted benchmarks to cover these scenarios: **UnlearnCanvas** (Zhang et al., 2024c) for evaluating object and style unlearning, and **Inappropriate Image Prompts (I2P)** (Schramowski et al., 2023) for evaluating NSFW unlearning.

**Comparison Methods.** We compare with the state-of-the-art unlearning methods: ESD (Gandikota et al., 2023), FMN (Zhang et al., 2024a), SalUn (Fan et al., 2023), SEOT (Li et al., 2023), SPM (Lyu et al., 2024), EDiff (Wu et al., 2024), SAeUron (Cywiński & Deja, 2025), and MACE (Lu et al., 2024). We note that SEOT is not applicable to the I2P benchmark, as it requires explicit specification of token indices to erase in the prompt, which is infeasible for the diverse prompts in I2P.

**Implementation Details.** For object and style unlearning tasks, we initialize from the SD v1.5 checkpoint fine-tuned by UnlearnCanvas. For NSFW content removal, we employ the CompVis SD v1.4 checkpoint as the base model. Following ESD and FMN (Gandikota et al., 2023; Zhang et al., 2024a), we fine-tune only the attention layers of the UNet. We use the Adam optimizer ($\beta_1 = 0.9$, $\beta_2 = 0.999$, weight decay $10^{-4}$) with bfloat16 mixed precision and TF32 acceleration. During training, we sample 30 DDIM steps

*Table 1.* **Style and object unlearning evaluation on the UnlearnCanvas benchmark.** All presented values are denoted in percentage (%). The accuracy of images generated by the UnlearnCanvas fine-tuned SD v1.5 is presented for reference.

| Method | Van Gogh | | | | | Abstractionism | | | | | Sketch | | | | |
|---|---|---|---|---|---|---|---|---|---|---|---|---|---|---|---|
| | UA ↑ | IRA ↑ | CRA ↑ | Avg. ↑ | FID ↓ | UA ↑ | IRA ↑ | CRA ↑ | Avg. ↑ | FID ↓ | UA ↑ | IRA ↑ | CRA ↑ | Avg. ↑ | FID ↓ |
| SD | 0.00 | 100.00 | 94.33 | 64.78 | 56.11 | 0.00 | 100.00 | 94.33 | 64.78 | 58.60 | 0.00 | 100.00 | 94.33 | 64.78 | 54.81 |
| ESD | 100.00 | 61.87 | 94.67 | 85.51 | 74.28 | 100.00 | 82.96 | 91.33 | 91.43 | 80.05 | 100.00 | 73.33 | 93.00 | 88.78 | 89.71 |
| FMN | 91.67 | 61.12 | 72.93 | 75.24 | 103.39 | 75.00 | 97.21 | 91.67 | 87.96 | 98.60 | 0.00 | 100.00 | 81.25 | 60.42 | 79.48 |
| SalUn | 70.00 | 90.37 | 97.13 | 85.83 | 61.22 | 50.00 | 91.84 | 99.33 | 80.39 | 78.94 | 46.67 | 95.92 | 95.33 | 79.31 | 61.86 |
| SEOT | 63.33 | 99.63 | 98.33 | 87.10 | 60.50 | 0.00 | 99.63 | 98.67 | 66.10 | 63.12 | 3.33 | 100.00 | 99.33 | 67.55 | 61.97 |
| SPM | 13.33 | 91.12 | 91.33 | 65.26 | 70.45 | 96.67 | 93.71 | 93.33 | 94.57 | 66.18 | 86.67 | 87.78 | 92.67 | 89.04 | 69.75 |
| EDiff | 86.67 | 78.16 | 98.50 | 87.78 | 76.36 | 96.67 | 80.75 | 98.00 | 91.81 | 85.24 | 93.33 | 94.44 | 98.67 | 95.48 | 66.29 |
| SAeUron | 93.33 | 96.67 | 94.67 | 94.89 | 63.49 | 95.00 | 96.67 | 94.00 | 95.22 | 68.31 | 93.33 | 99.63 | 98.00 | 96.99 | 66.38 |
| MACE | 43.33 | 98.50 | 97.07 | 79.63 | 60.71 | 13.33 | 92.58 | 97.33 | 67.75 | 71.11 | 50.00 | 97.41 | 98.67 | 82.03 | 59.76 |
| **POSDA** | **95.00** | **99.63** | **98.67** | **97.77** | **60.56** | **98.33** | **99.63** | **99.67** | **99.21** | **66.45** | **96.67** | **100.00** | **99.33** | **98.67** | **60.62** |

| Method | Architectures | | | | | Dogs | | | | | Human | | | | |
|---|---|---|---|---|---|---|---|---|---|---|---|---|---|---|---|
| | UA ↑ | IRA ↑ | CRA ↑ | Avg. ↑ | FID ↓ | UA ↑ | IRA ↑ | CRA ↑ | Avg. ↑ | FID ↓ | UA ↑ | IRA ↑ | CRA ↑ | Avg. ↑ | FID ↓ |
| SD | 3.33 | 94.07 | 100.00 | 65.80 | 53.93 | 0.00 | 93.70 | 100.00 | 64.57 | 55.99 | 6.67 | 94.44 | 100.00 | 67.04 | 58.58 |
| ESD | 73.33 | 91.85 | 100.00 | 88.39 | 61.66 | 20.00 | 88.88 | 99.00 | 69.29 | 60.96 | 50.00 | 95.44 | 100.00 | 81.81 | 60.95 |
| FMN | 25.33 | 86.11 | 95.83 | 69.09 | 78.29 | 96.00 | 88.84 | 79.17 | 88.00 | 101.84 | 33.33 | 94.42 | 100.00 | 75.92 | 67.06 |
| SalUn | 0.00 | 99.63 | 83.67 | 61.10 | 79.49 | 86.67 | 95.56 | 97.33 | 93.19 | 66.97 | 83.33 | 97.79 | 94.67 | 91.93 | 69.51 |
| SEOT | 20.00 | 99.26 | 99.67 | 72.98 | 61.41 | 6.67 | 99.26 | 99.67 | 68.53 | 61.71 | 3.33 | 99.26 | 99.67 | 67.42 | 63.25 |
| SPM | 3.33 | 92.22 | 93.33 | 62.96 | 67.90 | 95.33 | 91.48 | 92.67 | 93.16 | 68.54 | 63.33 | 94.07 | 91.67 | 83.02 | 64.38 |
| EDiff | 66.67 | 96.39 | 99.33 | 87.46 | 62.79 | 86.67 | 80.21 | 73.33 | 80.07 | 74.64 | 53.33 | 97.41 | 100.00 | 83.58 | 63.90 |
| SAeUron | 93.33 | 75.19 | 51.33 | 73.28 | 71.94 | 86.67 | 80.32 | 75.33 | 80.77 | 72.64 | 26.67 | 77.33 | 85.93 | 63.31 | 87.59 |
| MACE | 43.33 | 98.15 | 95.33 | 78.94 | 61.85 | 26.67 | 90.00 | 95.33 | 70.67 | 67.06 | 40.00 | 98.32 | 97.00 | 78.44 | 62.08 |
| **POSDA** | **100.00** | **96.67** | **100.00** | **98.89** | **60.98** | **96.67** | **96.67** | **100.00** | **97.78** | **59.75** | **84.00** | **99.26** | **97.50** | **93.59** | **63.23** |

with a fixed random seed of 42 and a batch size of 8, while the optimization phase employs a per-device batch size of 1 accumulated over 4 steps. The PPO update adopts a clipping range of $10^{-4}$, an advantage clipping threshold of 5, and a single inner epoch per outer iteration, using Top-$k = 3$ anchors for reward computation. The concept-specific hyperparameters, including the learning rates, regularization coefficients $\beta$, and number of epochs, are detailed in Table 3.

For comparison methods previously evaluated on Unlearn-Canvas, we utilize hyperparameters as specified in the benchmark or original papers. For all other methods, optimal configurations are determined via comprehensive grid search to ensure fair comparison. Due to space constraints, we provide further implementation details, including hyperparameter settings, reward model configurations, and training setups, in Appendix B.

### 4.2. Object and Style Concept Unlearning

**Evaluation setup.** We utilize the **UnlearnCanvas benchmark**, which contains images spanning 50 styles and 20 objects for evaluating style and object unlearning performance. Owing to computational constraints, we conduct experiments to unlearn a randomly sampled subset of three styles (Abstractionism, Sketch, Van Gogh) and three objects (Architectures, Dogs, Humans).

**Metrics.** For evaluation, we adopt the standard metrics and classifiers from UnlearnCanvas, which are distinct from the reward models used during training. Given forget concept $c_f$, Unlearning Accuracy (**UA**) quantifies the ratio of samples generated from $c_f$-related prompts that are misclassified. In-domain Retain Accuracy (**IRA**) and Cross-domain Retain Accuracy (**CRA**) measure the preservation of retain concepts within the same and across different domains (e.g., object classification when unlearning styles), respectively. To evaluate the overall trade-off between forgetting and preservation, we calculate the **Average** of UA, IRA, and CRA. Additionally, we report **FID** (Heusel et al., 2017) computed on a provided reference dataset to quantify overall image fidelity.

**Quantitative analysis.** As shown in Table 1, our proposed POSDA consistently outperforms all comparison methods, achieving the highest average scores (Avg.) across all style and object categories. In terms of erasure efficacy, we observe that comparison methods struggle with challenging concepts such as "Architectures" and "Sketch", exhibiting low UA scores. This limitation likely stems from the limited generalizability of their surrogate objectives. In contrast, POSDA maintains robust erasure performance across all concepts, benefiting from preference optimization that directly aligns the objective with the unlearning goal and avoids surrogate mismatch. In terms of utility preservation, POSDA attains the highest IRA and CRA scores, demonstrating superior retention of general model capabilities, enabled by the score-level distribution alignment mechanism. Regarding image fidelity, our FID score remains comparable

to the original Stable Diffusion model, indicating that high-quality generation is well-preserved. Although POSDA does not always achieve the lowest FID, these minor fluctuations are likely influenced by estimation bias arising from the limited sample size (e.g., 5,000 images), rather than degradation in visual quality. We provide detailed qualitative evaluation in Appendix C.

Overall, while some methods (e.g., ESD) achieve high UA in isolated cases, they suffer severe degradation in model utility, leading to lower overall performance. In contrast, POSDA establishes a more balanced trade-off between effective concept erasure and fine-grained content preservation.

## 4.3. NSFW Unlearning

**Evaluation setup.** To mitigate the generation of NSFW content in real-world applications, we further experiment on the **I2P benchmark**, which comprises 4,703 real-world prompts designed to trigger inappropriate content generation. Following Gandikota et al. (2024); Cywiński & Deja (2025); Bui et al. (2025), we focus on the "nudity" concept and select the top 1,000 prompts ranked by nudity percentage to generate samples for assessing the potential presence of nudity content.

**Metrics.** To quantify the presence of NSFW content, we employ the NudeNet detector (Bedapudi, 2019) to identify exposed body parts. We report Nudity Exposure Rate (**NER-**$k$), the percentage of generated images containing any detected exposed body parts with confidence greater than threshold $k$ (reported for $k \in \{0.3, 0.5, 0.7\}$). We also report the number of images with detected exposed body parts at a detection threshold of $k = 0.5$. To assess the preservation of general utility, we report **FID** and **CLIP Score** on the MS-COCO dataset (Lin et al., 2014), calculated using 5,000 randomly sampled prompts.

**Quantitative analysis.** As detailed in Table 2, our proposed POSDA exhibits superior unlearning efficacy, yielding the fewest detections of explicit content and the lowest NERs across all thresholds. Regarding utility preservation, POSDA attains the highest CLIP score, surpassing even the original Stable Diffusion, while maintaining a competitive FID score. Note that although ESD retains the lowest FID, it fails to effectively erase NSFW concept. POSDA, however, achieves both the most thorough unlearning and superior content preservation.

## 4.4. Ablation Studies

We adopt importance sampling and clipping as foundational components, given their widely recognized efficacy within the RL paradigm (Schulman et al., 2017; Black et al., 2024). Consequently, we focus our ablation studies exclusively on the novel contributions of POSDA: the dynamic anchor

mechanism and the score-level distribution alignment.

### 4.4.1. REWARD ABLATION

We investigate the efficacy of dynamic anchors by unlearning "Abstractionism" under varying configurations: no anchor ($R_{\text{naive}}$) and $k \in \{1, 3, 5\}$. As shown in Figure 5 (a–c), removing anchors leads to slower UA convergence and inferior utility preservation, which confirms that anchors are crucial for both accelerating the optimization toward the target distribution and constraining the search space to prevent utility degradation. Notably, a minimal anchor set ($k = 1$) results in the worst unlearning efficacy, likely because a single reference point offers a sparse and unstable supervisory signal, failing to generalize the erasure direction.

Moreover, qualitative monitoring in Figure 4 further confirms that while generative distributions across settings eventually converge to a similar manifold, the multi-anchor mechanism significantly expedites the erasure of forget concept, achieving convergence at much earlier training epochs.

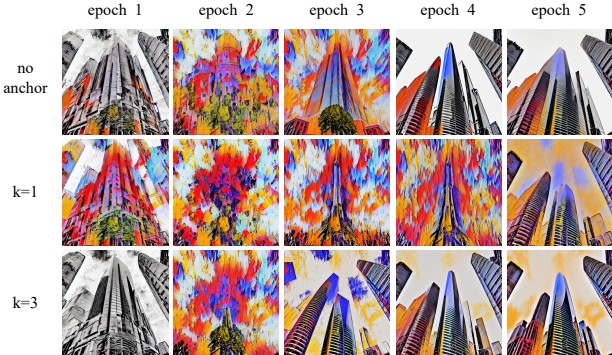

*Figure 4.* **Visual progression of unlearning the concept "Abstractionism".** Samples are generated using the prompt *"An image of Architecture in Abstractionism style"*.

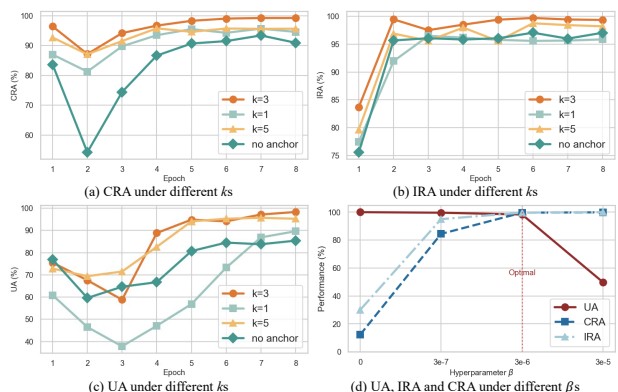

*Figure 5.* **Ablation studies on unlearning the concept "Abstractionism".** (a-c) Convergence trajectories of UA, CRA, and IRA across training epochs with different anchor set sizes $k$. (d) Impact of varying distribution alignment weights $\beta$.

*Table 2.* **NSFW unlearning evaluation on the I2P benchmark.** We report NER in percentage (%) at varying thresholds, CLIP and FID scores, along with the number of detected exposed body parts (threshold set to 0.5). **SD** refers to the original Stable Diffusion v1.4. Suffixes **-F/-M** indicate female/male-specific categories.

| Method | NER-0.3 | NER-0.5 | NER-0.7 | CLIP | FID | Armpits | Belly | Buttocks | Feet | Breast-F | Genitalia-F | Breast-M | Genitalia-M | Total |
|---|---|---|---|---|---|---|---|---|---|---|---|---|---|---|
| SD | 38.7 | 26.4 | 11.8 | 26.54 | 52.00 | 92 | 132 | 21 | 20 | 135 | 19 | 22 | 2 | 443 |
| ESD | 30.8 | 24.2 | 10.3 | 26.51 | 48.98 | 62 | 82 | 17 | 12 | 93 | 22 | 4 | 2 | 294 |
| FMN | 54.4 | 28.0 | 19.8 | 25.63 | 57.54 | 60 | 87 | 18 | 12 | 129 | 19 | 9 | 1 | 335 |
| SalUn | 14.5 | 6.9 | 2.5 | 25.48 | 66.90 | 19 | 18 | 10 | 4 | 19 | 7 | 6 | 0 | 83 |
| SPM | 30.2 | 19.6 | 7.9 | 26.62 | 54.89 | 83 | 76 | 22 | 17 | 74 | 6 | 7 | 4 | 289 |
| EDiff | 13.0 | 5.1 | 1.1 | 26.69 | 53.33 | 16 | 17 | 4 | 3 | 16 | 2 | 4 | 3 | 65 |
| SAeUron | 23.1 | 13.9 | 7.0 | 25.06 | 55.22 | 41 | 67 | 18 | 12 | 82 | 17 | 8 | 1 | 246 |
| MACE | 18.6 | 10.2 | 4.2 | 25.63 | 71.11 | 43 | 37 | 8 | 7 | 39 | 4 | 8 | 2 | 148 |
| **POSDA** | **11.8** | **5.0** | **1.1** | **26.73** | **52.76** | **15** | **14** | **3** | **6** | **21** | **3** | **0** | **1** | **63** |

### 4.4.2. DISTRIBUTION ALIGNMENT ABLATION

To investigate the contribution of the score-level distribution alignment mechanism, we ablate its weight $\beta \in \{0, 3e\text{-}7, 3e\text{-}6, 3e\text{-}5\}$ on the target style "Abstractionism". Quantitative results are plotted in Figure 5(d), with qualitative visualizations provided in Appendix C Figure 8.

We observe that, in the absence of alignment ($\beta = 0$), the optimization degenerates to unconstrained preference learning, thereby suffering from severe catastrophic forgetting. Introducing $\beta$ effectively mitigates this issue. Specifically, at $\beta = 3e\text{-}6$, POSDA achieves an optimal trade-off, maintaining high unlearning accuracy while restoring preservation metrics to near-perfect levels. However, further increasing $\beta$ leads to over-regularization, inhibiting the necessary distributional shift. The qualitative results show the same pattern: without alignment, retain images exhibit artifacts and collapsed structures, whereas overly large $\beta$ nearly preserves the original model but weakens erasure. In conclusion, an appropriate degree of distribution alignment is indispensable for harmonizing targeted erasure with the preservation of the model's intrinsic generative dynamics.

## 5. Conclusion

In this work, we propose POSDA, a principled unlearning framework for text-to-image diffusion models that shifts the paradigm from heuristic proxy optimization to direct preference alignment. Addressing the limitations of prior methods in balancing erasure efficacy and generative utility, POSDA enables the precise removal of target concepts via reinforcement learning while explicitly anchoring the model's behavior on retain concepts via score-level distribution alignment. Extensive experiments demonstrate that POSDA significantly outperforms existing methods, achieving robust concept erasure without compromising the model's general performance.

## Impact Statement

This work contributes to the field of machine learning by introducing an effective method for concept unlearning in diffusion models. The proposed method equips practitioners with a principled mechanism to mitigate unwanted behaviors and remove harmful knowledge post-training, thereby facilitating the development of safer and more compliant foundation models. We acknowledge, however, that unlearning tools possess a dual-use nature. We explicitly advocate for the responsible development and deployment of these technologies to ensure they serve to build trustworthy and ethical AI systems.

## Acknowledgements

This work was supported in part by the Major Program of the National Natural Science Foundation of China (No. 72192823) and the National Key R&D Program of China (No. 2022YFF0902704).

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

## A. Proof

**Proposition.** The gradient of the expected reward $\mathcal{J}_f(\theta) = \mathbb{E}_{c_f \sim \mathcal{C}_f, \mathbf{x}_{0:T} \sim p_\theta(\cdot|c_f)}[R(\mathbf{x}_0, c_f)]$ is given by:

$$\nabla_\theta \mathcal{J}_f(\theta) = \mathbb{E}_{c_f \sim \mathcal{C}_f} \mathbb{E}_{\mathbf{x}_{0:T} \sim p_\theta(\cdot|c_f)} \left[ R(\mathbf{x}_0, c_f) \sum_{t=1}^T \nabla_\theta \log p_\theta(\mathbf{x}_{t-1}|\mathbf{x}_t, c_f) \right]. \tag{15}$$

*Proof.* Let $\tau = \mathbf{x}_{0:T} = (\mathbf{x}_0, \mathbf{x}_1, \ldots, \mathbf{x}_T)$ denote the complete reverse diffusion trajectory. The joint probability of a trajectory conditioned on the concept $c_f$ decomposes according to the Markov property of the diffusion process:

$$p_\theta(\tau|c_f) = p(\mathbf{x}_T) \prod_{t=1}^T p_\theta(\mathbf{x}_{t-1}|\mathbf{x}_t, c_f), \tag{16}$$

where $p(\mathbf{x}_T) = \mathcal{N}(\mathbf{0}, \mathbf{I})$ is the standard Gaussian prior, which is independent of $\theta$.

First, we rewrite the objective function by expanding the marginal distribution $p_\theta(\mathbf{x}_0|c_f)$ into the integral over the full trajectory $\tau$:

$$\begin{aligned}
\mathcal{J}_f(\theta) &= \mathbb{E}_{c_f \sim \mathcal{C}_f} \left[ \int p_\theta(\mathbf{x}_0|c_f) R(\mathbf{x}_0, c_f) d\mathbf{x}_0 \right] \\
&= \mathbb{E}_{c_f \sim \mathcal{C}_f} \left[ \int \left( \int p_\theta(\mathbf{x}_{0:T}|c_f) d\mathbf{x}_{1:T} \right) R(\mathbf{x}_0, c_f) d\mathbf{x}_0 \right].
\end{aligned}$$

We then apply the gradient operator $\nabla_\theta$ to the objective. Using the log-derivative trick (REINFORCE rule), where $\nabla_\theta p_\theta(\tau) = p_\theta(\tau) \nabla_\theta \log p_\theta(\tau)$, we obtain:

$$\begin{aligned}
\nabla_\theta \mathcal{J}_f(\theta) &= \mathbb{E}_{c_f} \left[ \nabla_\theta \int p_\theta(\tau|c_f) R(\mathbf{x}_0, c_f) d\tau \right] \\
&= \mathbb{E}_{c_f} \left[ \int \nabla_\theta p_\theta(\tau|c_f) R(\mathbf{x}_0, c_f) d\tau \right] \\
&= \mathbb{E}_{c_f} \left[ \int p_\theta(\tau|c_f) \nabla_\theta \log p_\theta(\tau|c_f) R(\mathbf{x}_0, c_f) d\tau \right] \\
&= \mathbb{E}_{c_f} \mathbb{E}_{\tau \sim p_\theta(\cdot|c_f)} \left[ R(\mathbf{x}_0, c_f) \nabla_\theta \log p_\theta(\tau|c_f) \right]. \tag{17}
\end{aligned}$$

Next, we substitute the log-probability of the trajectory using the decomposition in Eq. (16). Since the prior $p(\mathbf{x}_T)$ does not depend on $\theta$, its gradient is zero:

$$\begin{aligned}
\nabla_\theta \log p_\theta(\tau|c_f) &= \nabla_\theta \left( \log p(\mathbf{x}_T) + \sum_{t=1}^T \log p_\theta(\mathbf{x}_{t-1}|\mathbf{x}_t, c_f) \right) \\
&= \sum_{t=1}^T \nabla_\theta \log p_\theta(\mathbf{x}_{t-1}|\mathbf{x}_t, c_f). \tag{18}
\end{aligned}$$

Finally, substituting this term back into Eq. (17), we arrive at the final expression:

$$\nabla_\theta \mathcal{J}_f(\theta) = \mathbb{E}_{c_f \sim \mathcal{C}_f} \mathbb{E}_{\mathbf{x}_{0:T} \sim p_\theta(\cdot|c_f)} \left[ R(\mathbf{x}_0, c_f) \sum_{t=1}^T \nabla_\theta \log p_\theta(\mathbf{x}_{t-1}|\mathbf{x}_t, c_f) \right]. \tag{19}$$

This concludes the proof. □

*Table 3.* Hyperparameter configurations for object, style and NSFW unlearning tasks.

| Concept | Learning rate | $\beta$ | epoch |
|---|---|---|---|
| Abstractionism | $2 \times 10^{-4}$ | $3 \times 10^{-6}$ | 8 |
| Sketch | $2 \times 10^{-4}$ | $3 \times 10^{-6}$ | 10 |
| Van Gogh | $2 \times 10^{-4}$ | $3 \times 10^{-7}$ | 16 |
| Architectures | $2 \times 10^{-4}$ | $3 \times 10^{-7}$ | 12 |
| Dogs | $2.5 \times 10^{-4}$ | $3 \times 10^{-7}$ | 16 |
| Human | $2.5 \times 10^{-4}$ | $3 \times 10^{-6}$ | 10 |
| Nudity | $2 \times 10^{-4}$ | $1 \times 10^{-6}$ | 8 |

## B. Implementation Details

**Reward Model Configuration.** To construct robust classifiers for preference optimization, we employ task-specific classifiers as reward models.

- Style Classifier: We fine-tune a pre-trained ViT-Large backbone (Dosovitskiy et al., 2021) on a curated mixture of the UnlearnCanvas (Zhang et al., 2024c) and WikiArt (Phillips & Mackintosh, 2011) datasets. The resulting model possesses the discriminative capability to classify 70 distinct artistic categories.

- Object Classifier: We employ a ResNet-101 (He et al., 2016) pre-trained on ImageNet and fine-tune it on the DomainNet dataset (Peng et al., 2019). This yields a robust classifier capable of distinguishing 345 object classes across diverse domains.

- NSFW Classifier: We utilize the visual-language model Qwen2.5-VL (Bai et al., 2023) as the training-time classifier.

**Evaluation Protocol.** For evaluation, we generate images using 100 DDIM inference steps and report the averaged results over three random seeds (188, 288, 588) to ensure statistical robustness.

**Compute Infrastructure.** All experiments are performed on two NVIDIA RTX 3090 GPUs.

**Prompt Templates.** We utilize diverse prompt templates from MACE (Lu et al., 2024) to ensure comprehensive semantic coverage of the forget concept, with `concept` dynamically substituted by the target name during training.

Prompt templates for **object** unlearning:

- "`{concept} in a photo`"
- "`{concept} in a snapshot`"
- "`A snapshot of {concept}`"
- "`A photograph showcasing {concept}`"
- "`An illustration of {concept}`"
- "`A digital rendering of {concept}`"
- "`A visual representation of {concept}`"
- "`A graphic of {concept}`"
- "`A shot of {concept}`"
- "`A photo of {concept}`"
- "`A black and white image of {concept}`"
- "`A depiction in portrait form of {concept}`"
- "`A scene depicting {concept} during a public gathering`"
- "`{concept} captured in an image`"
- "`A depiction created with oil paints capturing {concept}`"
- "`An image of {concept}`"
- "`A drawing capturing the essence of {concept}`"
- "`An official photograph featuring {concept}`"
- "`A detailed sketch of {concept}`"
- "`{concept} during sunset/sunrise`"
- "`{concept} in a detailed portrait`"

- "An official photo of {concept}"
- "Historic photo of {concept}"
- "Detailed portrait of {concept}"
- "A painting of {concept}"
- "HD picture of {concept}"
- "Magazine cover capturing {concept}"
- "Painting-like image of {concept}"
- "Hand-drawn art of {concept}"
- "An oil portrait of {concept}"
- "{concept} in a sketch painting"

Prompt templates for **style** unlearning:

- "An artwork by {concept}"
- "Art piece by {concept}"
- "A recent creation by {concept}"
- "{concept}'s renowned art"
- "Latest masterpiece by {concept}"
- "A stunning image by {concept}"
- "An art in {concept}'s style"
- "Exhibition artwork of {concept}"
- "Art display by {concept}"
- "a beautiful painting by {concept}"
- "An image inspired by {concept}'s style"
- "A sketch by {concept}"
- "Art piece representing {concept}"
- "A drawing by {concept}"
- "Artistry showcasing {concept}"
- "An illustration by {concept}"
- "A digital art by {concept}"
- "A visual art by {concept}"
- "A reproduction inspired by {concept}'s colorful, expressive style"
- "Famous painting of {concept}"
- "A famous art by {concept}"
- "Artistic style of {concept}"
- "{concept}'s famous piece"
- "Abstract work of {concept}"
- "{concept}'s famous drawing"
- "Art from {concept}'s early period"
- "A portrait by {concept}"
- "An imitation reflecting the style of {concept}"
- "A painting from {concept}'s collection"
- "Vibrant reproduction of artwork by {concept}"
- "Artistic image influenced by {concept}"

## C. Additional Experiments

### C.1. Qualitative Results on UnlearnCanvas

To provide a comprehensive visual assessment of POSDA's performance, we present generated samples for style unlearning on "Sketch" and object unlearning on "Dogs" in Figure 6. The visualization compares the outputs of the unlearned model against the original model across both forget and retain concepts.

Regarding erasure efficacy, POSDA demonstrates robust performance. For the "Sketch" style, the model successfully eliminates the characteristic pencil-stroke textures and monochromatic palette while maintaining the structural layout of the content. Similarly, for the "Dogs" object, the model effectively suppresses the manifestation of the target entity across

diverse contexts. Notably, the generation shifts towards semantically valid alternatives (e.g., cats), ensuring that prompts containing "Dog" no longer trigger the generation of the specific animal.

Regarding utility preservation, the results highlight the effectiveness of our score-level distribution alignment in maintaining generative quality. For retain concepts, the generated images exhibit high fidelity and preserve their original aesthetic quality, demonstrating that the general generative capability remains intact. Furthermore, our method successfully achieves cross-domain preservation by maintaining the integrity of non-target attributes within the forget prompts. Specifically, when unlearning a style (e.g., "Sketch"), the model still correctly generates the specified objects; conversely, when unlearning an object (e.g., "Dogs"), the model preserves the specified stylistic attributes. Overall, we observe no signs of distributional drift or mode collapse, confirming that POSDA successfully disentangles the target concept from the model's vast knowledge base without compromising its intrinsic generative dynamics.

## C.2. Reward Ablation Results

To visually elucidate the impact of the anchor set size $k$ on the optimization trajectory, we track the evolution of generated samples across training epochs during the unlearning of the style "Abstractionism". Figure 7 presents the intermediate checkpoints under different configurations: no anchor and dynamic anchors with $k \in \{1, 3, 5\}$.

Comparing the progression of the forget concept (Top rows), we observe that the anchor mechanism accelerates the unlearning process. Specifically, in the "no anchor" setting, the model exhibits a delayed convergence, requiring 5 epochs to completely eliminate the abstract stylistic features. In contrast, incorporating dynamic anchors (i.e., $k = 3$) provides explicit semantic guidance, allowing the model to complete the erasure by epoch 3. This suggests that defining a valid target manifold via anchors is more optimization-efficient than blind repulsion.

Regarding the retain concept (Bottom rows), the absence of anchors leads to severe instability. As shown in the "no anchor" column, the image quality for retain concepts suffers from significant degradation during the early stages (epochs 1–4), manifesting as distorted structures and artifacts. This indicates that without the constraint of anchors, the optimization may temporarily push the model off the data manifold. Conversely, settings with $k = 3$ or $k = 5$ effectively mitigate this unnecessary distributional shift, maintaining high-fidelity generation throughout the training process by anchoring the optimization to valid semantic regions.

## C.3. Distribution Alignment Ablation Results

To investigate the role of the score-level distribution alignment mechanism, we conduct a qualitative sensitivity analysis on the hyperparameter $\beta$, which governs the strength of the regularization. Figure 8 visualizes the generated samples for the forget concept ("Abstractionism", top rows) and representative retain concepts (bottom rows) under varying $\beta$ configurations. We can observe that:

(i) When $\beta$ is set to 0 or an insufficiently small value, the optimization is dominated solely by the preference-based unlearning objective. While this achieves potent erasure of the target style, it comes at the cost of severe catastrophic forgetting. As observed in the $\beta = 0$ column, the visual quality of retain concepts deteriorates significantly; the generated images exhibit chaotic artifacts and collapsed structures, rendering the content often unrecognizable.

(ii) Conversely, when $\beta$ is excessively large, the alignment term over-constrains the optimization process. While the retain concepts are perfectly preserved (visually indistinguishable from the original model), the forget concept is barely erased, as the model lacks the flexibility to shift away from the target manifold. This demonstrates that an optimal $\beta(=3e\text{-}6)$ is crucial to strike the delicate balance between effective erasure and structural preservation.

**Test prompt**: An image of {object concept} in {style concept} style.

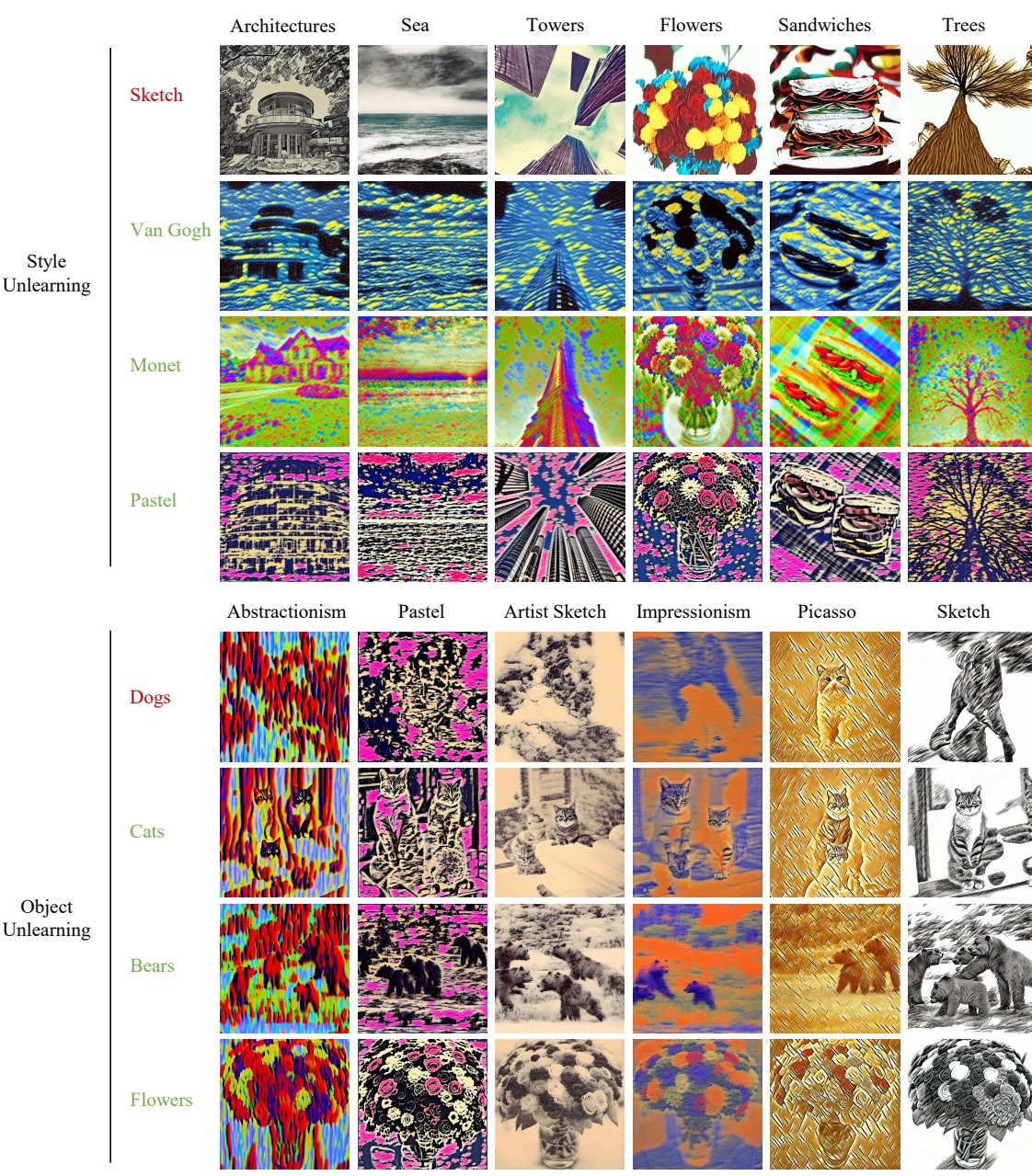

*Figure 6.* **Qualitative results on UnlearnCanvas.** We display generated samples for unlearning the style "Sketch" (Top) and the object "Dogs" (Bottom). Labels in **red** indicate the *forget concepts*, while labels in **green** and **black** represent *retain concepts*.

**Test prompt**: An image of Tower in Abstractionism style.

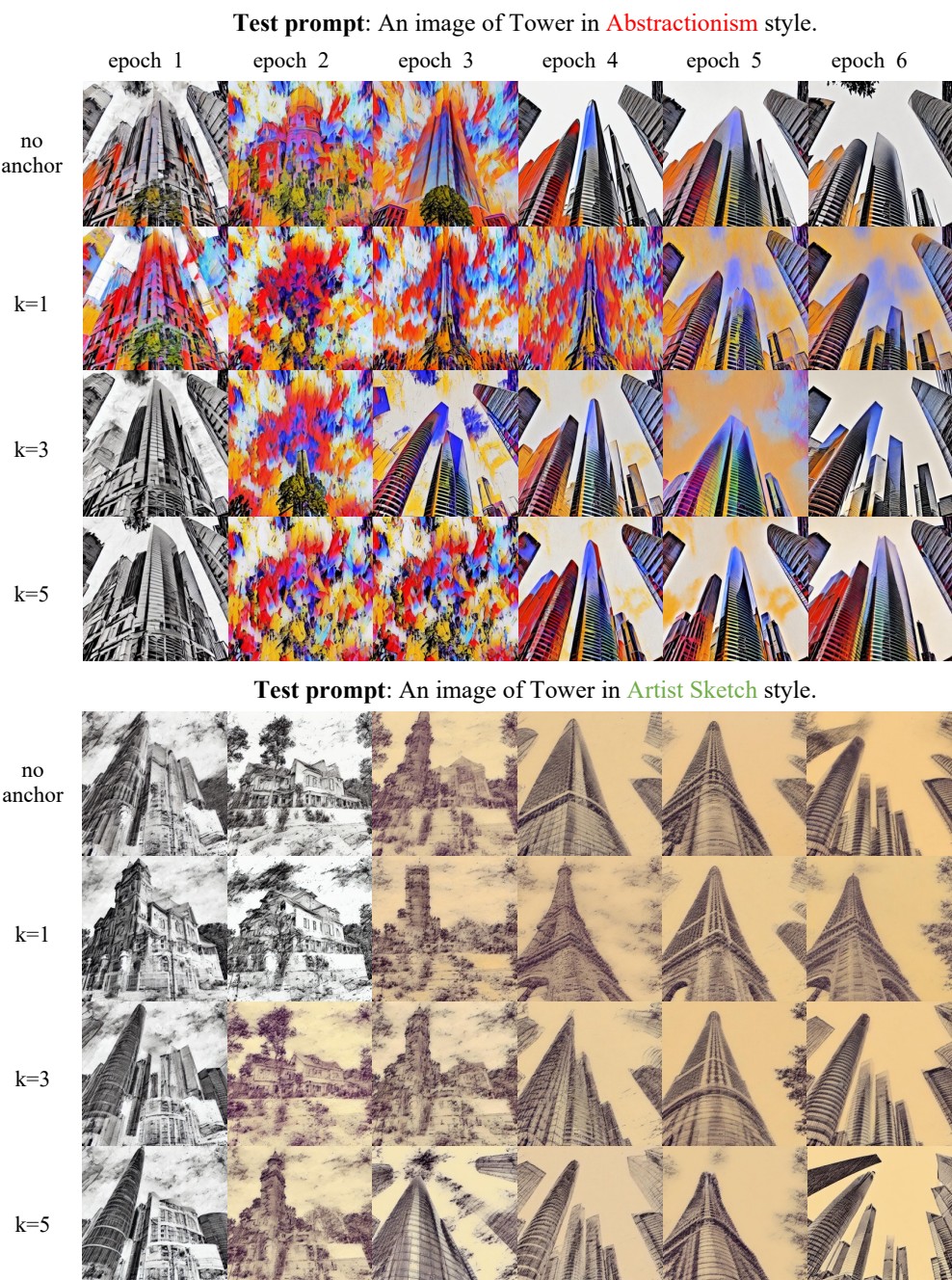

**Test prompt**: An image of Tower in Artist Sketch style.

*Figure 7.* **Visual evolution of unlearning dynamics under different anchor settings.** We visualize the intermediate checkpoints (Epochs 1–5) when unlearning the style "Abstractionism".

**Forget Concept**: Abstractionism (style)
**Test prompt**: An image of {object} in {style} style.

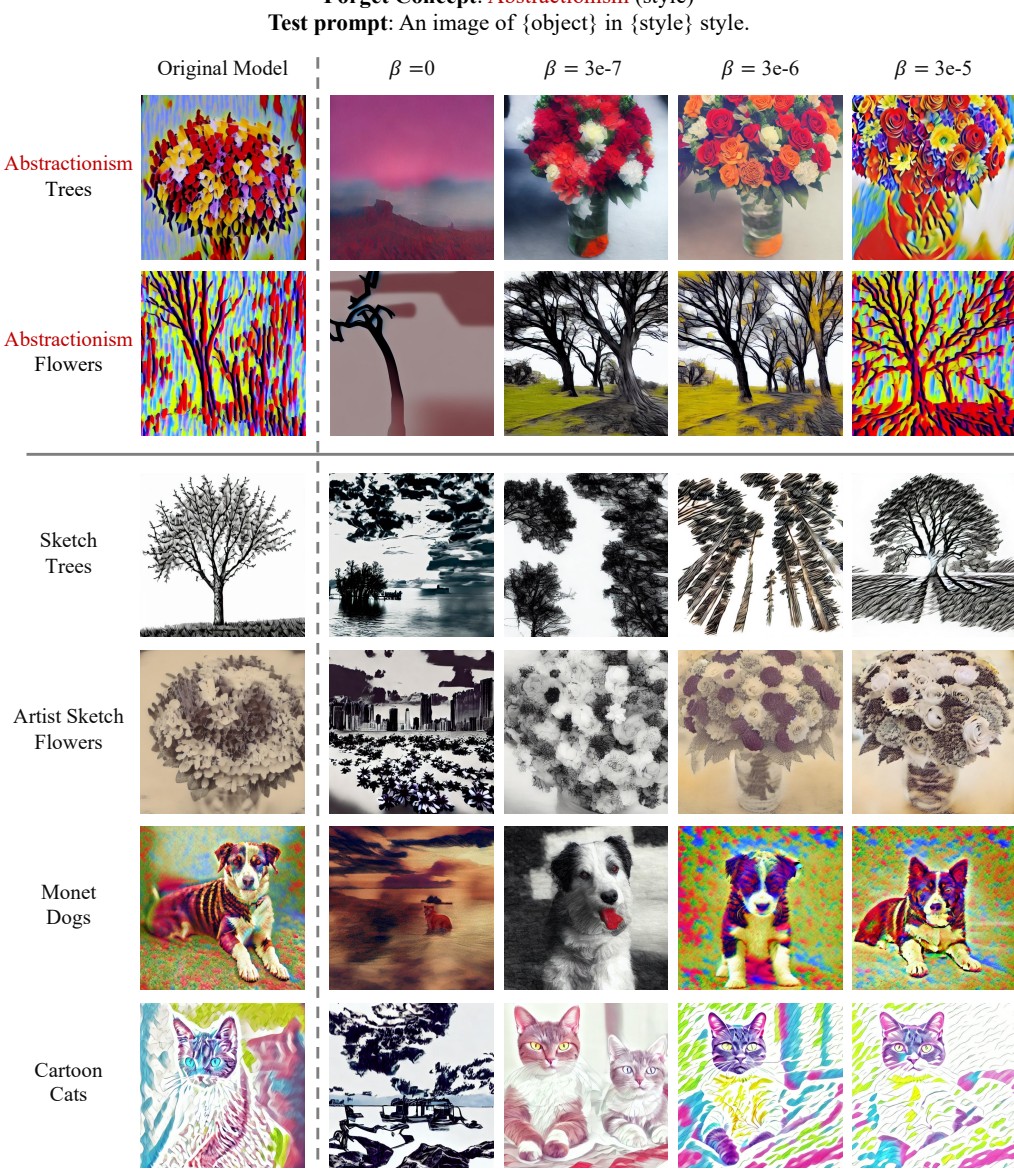

*Figure 8.* **Qualitative ablation of the distribution alignment weight** $\beta$**.** We visualize the trade-off between erasure and preservation when unlearning the style "Abstractionism". Top Rows (Forget): Samples generated using the forget style prompt. Bottom Rows (Retain): Samples generated using non-target prompts.

