# OpenReview forum: "Preference-Calibrated Optimization with Score-Level Distribution Alignment for Text-to-Image Diffusion Model Unlearning"
_ICML.cc/2026/Conference — ICML 2026 regular_

### Official Review · Reviewer_s2Lh · 2026-03-09

**Soundness:** 1
**Presentation:** 2
**Significance:** 2
**Originality:** 3
**Overall Recommendation:** 4
**Confidence:** 4

**Summary:**

This paper tackles the problem of machine unlearning in text-to-image diffusion models. Existing methods often rely on handcrafted surrogate objectives that fail to robustly erase target concepts, and they apply surface-level constraints that lead to catastrophic forgetting of non-target concepts. To address these issues, the authors propose POSDA (Preference-Calibrated Optimization with Score-Level Distribution Alignment), a novel framework comprising two main components. First, it reframes unlearning as a reinforcement learning-based preference optimization problem, utilizing a dynamic anchor-based reward to directly steer the generation away from the forget concept toward valid semantic neighbors. Second, it introduces a score-level distribution alignment mechanism to minimize the KL divergence between the unlearned and original models on retained concepts, thereby maintaining intrinsic generative dynamics and mitigating catastrophic forgetting. Extensive experiments on object, style, and NSFW unlearning benchmarks demonstrate that POSDA achieves a strong balance between targeted concept erasure and general utility preservation compared to state-of-the-art baselines.

**Compliance With Llm Reviewing Policy:**

Affirmed.

**Final Justification:**

The author clarified some misunderstandings caused by concise derivation, and have committed to revising the derivation process and improving notational rigor in future versions of the paper.  Therefore, I give it 4

**Key Questions For Authors:**

1. **Details on Qwen2.5-VL Classifier Probabilities:** When handling the NSFW unlearning task, the authors employ Qwen2.5-VL as the reward classifier. For such an autoregressive vision-large language model, could you elaborate on how the continuous classification probability $P_{\phi}(c|x)$ is concretely calculated and extracted? For instance, do you extract the post-softmax logits of specific class label tokens, or do you employ a specific prompting mechanism? I recommend adding these crucial implementation details to the appendix to facilitate future reproducibility.

2. **Upper Bound Relationship between Equation 8 and Equation 9:** The paper posits that Equation 9 serves as an upper bound for Equation 8 (i.e., the KL divergence of the distributions in the final noise-free pixel space $x_0$). However, the manuscript does not provide a rigorous mathematical proof or relevant citations for this conclusion. Could you please provide a detailed explanation or proof of why the expected divergence of marginal distributions of noisy data across all intermediate timesteps necessarily constitutes an upper bound for the final generated distribution divergence?

3. **Equivalence of Multi-Step and Single-Step Gradients in Equation 13:** In deriving Equation 13, the main text states that $x_0$ is generated via a "deterministic sampling process" to ensure differentiability. Yet, in the subsequent chain rule application, the term $\frac{\partial x_0}{\partial \theta}$ is directly substituted with the single-step denoising mean gradient $\frac{\partial \mu_\theta}{\partial \theta}$ using a strict equality sign ($=$). Why is backpropagation through the entire multi-step sampling trajectory equivalent to a single-step gradient?

4. **Omission in the Derivative from Equation 9 to Equation 10:** In the process of taking the derivative of Equation 9 with respect to the parameter $\theta$ to derive Equation 10, a key term appears to be omitted. When applying a strict chain rule derivative to the new model term $\log p_{\theta,t}(x_t|c_r)$, in addition to the gradient term caused by the change in the input variable $x_t$ (which is the score-based term currently retained in Equation 10), there should logically be a direct partial derivative of the probability distribution network itself with respect to its parameters $\theta$ (i.e., the network Jacobian term $\frac{\partial \log p_{\theta,t}(x_t)}{\partial \theta}$). On what theoretical basis is this term omitted here? To ensure the rigor of the derivation, I recommend providing an explicit explanation and justification for omitting this term in the manuscript.

**Limitations:**

yes

**Strengths And Weaknesses:**

**Strengths:**

**Novel and Well-Motivated Perspective:** The paper astutely identifies the "objective mismatch" in existing unlearning methods that rely on heuristic surrogate losses. Reframing machine unlearning as a reinforcement learning-based preference optimization problem is an intuitive, original, and highly effective approach. Additionally, the introduction of the dynamic anchor mechanism to restrict unbounded repulsion is particularly clever and effectively enhances optimization stability.

**Extensive and Convincing Experiments:** The empirical validation is comprehensive. The authors evaluate the POSDA framework across diverse and challenging scenarios, including objects, artistic styles, and real-world NSFW content. Compared to a wide range of baselines (e.g., ESD, FMN, SalUn, MACE), POSDA consistently demonstrates state-of-the-art performance, achieving an excellent trade-off between targeted concept erasure and non-target image fidelity.



**Weaknesses:**

**Lack of Rigor and Formality in Mathematical Derivations:** Some mathematical derivation steps in the paper omit necessary theoretical justifications and exhibit logical leaps or omissions (specific details are elaborated in the "Key Questions for Authors" section). Furthermore, the mathematical notations lack rigor. For instance, in derivations involving vector-by-vector derivatives (such as the derivation process for Equation 13), the result should mathematically be a Jacobian matrix. However, the equations in the paper seem to treat and simplify these directly as scalars. Such informal notation significantly weakens the rigorousness of the theoretical derivations.

**Over-Reliance on High-Quality External Classifiers:** The proposed preference optimization relies heavily on task-specific classifiers (e.g., ViT-Large, ResNet-101, Qwen2.5-VL) to serve as reward models. However, the paper lacks a discussion on the framework's robustness: how would the unlearning efficacy be affected if the "target forget concept" is too niche to obtain a robust classifier, or if the external classifier itself suffers from inherent biases and recognition errors?

---

> ### Author Rebuttal · Authors · 2026-03-31
>
> We appreciate your thoughtful feedback, which were highly valuable in improving the quality of our paper. We have provided detailed clarifications for all the points you raised.
>
> **W1. Formality concerns.** We thank the reviewer for pointing this out. We will revise the manuscript to improve clarity by refining the notation, explicitly distinguishing vector and matrix quantities, and adding missing derivation details.
>
> **W2. Relationship with external classifiers.**
>
> Our method is agnostic to the specific choice of classifier. In practice, modern vision-language models provide strong zero-shot and fine-grained concept recognition capabilities, and can be directly used as reward models (e.g. Qwen2.5-VL for NSFW unlearning).
> For more niche concepts where zero-shot models are insufficient, lightweight classifiers can be trained with modest data. For style unlearning task, while CLIP struggles with fine-grained styles (as shown in Fig. 2), a fine-tuned classifier based on ViT enables effective unlearning in such cases.
>
> More broadly, reliance on a reward model is inherent and common for preference-based optimization methods. Our method makes this dependency explicit and modular, allowing the reward model to be improved or replaced as better models become available.
>
> **Q1. Clarification of Qwen2.5-VL reward construction.**
> In our NSFW experiments, Qwen2.5-VL functions not as a binary classifier, but as a fine-grained scorer that mimics NudeNet's taxonomy. Specifically, we prompt it to assign scores (0-10) to predefined exposed and clothed body parts. These scores are then normalized to compute $P_\phi(c\mid x)$. We will detail the procedure in the revised manuscript.
>
> **Q2. Clarification for Eq. 9**. We agree that describing Eq. (9) as a strict “upper bound” of Eq. (8) is imprecise, and we will revise this wording.
> The transition from Eq. (8) to Eq. (9) is instead a score-based approximate replacement. Directly optimizing the KL divergence in the clean data space (Eq. (8)) is intractable in our setting, since the distributions may have limited or non-overlapping support after unlearning, leading to ill-defined densities and unstable gradients.
>
> To address this, we adopt a common strategy in score-based generative modeling and variational score distillation [1], and optimize over noisy marginals across diffusion timesteps. Eq. (9) replaces the exact score in Eq. (8) with those computed on perturbed samples, and takes expectation over timesteps. The injected Gaussian noise acts as a smoothing mechanism that encourages distributional overlap and yields stable gradients.
> This results in a tractable and stable surrogate objective that preserves the alignment signal via score functions. We will revise the manuscript accordingly and remove the "upper bound" claim.
>
> [1] ProlificDreamer: High-fidelity and diverse text-to-3d generation with variational score distillation
>
> **Q3. Clarification for Eq.13**.
> Thank you for pointing this out. The notation $\frac{\partial x_0}{\partial \theta} = \frac{\partial \mu_\theta}{\partial \theta}$ in the current manuscript is not strictly accurate.
> In actual implementation, $x_0$ is obtained via a multi-step reverse diffusion process and the gradient $\frac{\partial x_0}{\partial \theta}$ is computed by automatic differentiation through the full sampling trajectory. More precisely, $x_0$ should be written as $x_0 = f_\theta(\xi, c_r)=\mu_\theta^{(1)}\circ \mu_\theta^{(2)}\circ \cdots \circ \mu_\theta^{(T)}(x_T,c_r,T)$, where $f_\theta$ denotes the full multi-step sampler. Accordingly, the correct gradient is $\frac{\partial x_0}{\partial \theta} = \frac{\partial f_\theta}{\partial \theta}$.
> The use of $\mu_\theta$ in Eq. (13) is a notational simplification to highlight the dependence on the denoising network, and should not be interpreted as implying equivalence between multi-step and single-step gradients. We will revise the manuscript to clarify this point and avoid confusion.
>
> **Q4. Justification for the derivative from Eq. 9 to Eq. 10.**
> Omitting $\frac{\partial \log p_{\theta,t}(x_t|c_r)}{\partial \theta}$ is a standard theoretical reduction, since its expectation under the model's own distribution evaluates strictly to zero:
> $\mathbb{E}_{x_t \sim p_{\theta,t}} \left[ \frac{\partial \log p_{\theta,t}(x_t)}{\partial \theta} \right] = \int p_{\theta,t}(x_t) \frac{\frac{\partial}{\partial \theta} p_{\theta,t}(x_t)}{p_{\theta,t}(x_t)} dx_t = \frac{\partial}{\partial \theta} \int p_{\theta,t}(x_t) dx_t = \frac{\partial}{\partial \theta} (1) = 0$
>
> Therefore, this term vanishes in expectation and does not contribute to the gradient. As a result, the remaining gradient is given by the pathwise term involving $\nabla_{x_t} \log p_{\theta,t}(x_t)$, which corresponds to the score function and leads to the form in Eq. (10). We will add a brief clarification in the manuscript to make this cancellation explicit and avoid confusion.

---

> > ### Author Rebuttal · Reviewer_s2Lh · 2026-04-02
> >
> > The authors have addressed my questions regarding the Qwen2.5-VL Classifier and Equation 10, and have committed to revising the derivation process and improving notational rigor in future versions of the paper. Based on these updates, I will raise my score to 4.

---

> > > ### Author Response · Authors · 2026-04-04
> > >
> > > Thank you very much for your positive feedback and careful re-evaluation of our rebuttal. We sincerely appreciate your time and constructive comments, which have greatly helped us improve our work.

---

### Official Review · Reviewer_5sed · 2026-03-11

**Soundness:** 4
**Presentation:** 3
**Significance:** 3
**Originality:** 3
**Overall Recommendation:** 4
**Confidence:** 4

**Summary:**

This paper proposes the POSDA framework, aiming to address two common pain points of diffusion models when performing the "machine forgetting" (Machine Unlearning) task: incomplete erasure due to mismatched target objectives, and "catastrophic forgetting" caused by ignoring the generation dynamics. Overall, the main idea of this manuscript is to achieve precise concept erasure without disrupting the underlying manifold structure of the model by reconfiguring the forgetting task as a preference optimization problem in reinforcement learning and combining score-level distribution alignment.

**Compliance With Llm Reviewing Policy:**

Affirmed.

**Final Justification:**

I appreciate the authors’ detailed rebuttal. The response partially addresses my concerns and improves several aspects of this paper, I will keep my score and don't have any other questions.

**Key Questions For Authors:**

See weakness above

**Limitations:**

Yes

**Strengths And Weaknesses:**

Strengths
1、The method is quite novel: The author did not use the simple noise mapping (Surrogate objectives), but instead transformed the forgetting task into preference optimization by constructing dynamic anchors. This approach is more in line with the essence of "forgetting" in theory.
2、The dynamic anchor mechanism is quite interesting: Unlike MACE or ESD which use fixed "neutral" words, POSDA automatically finds semantic neighbors as anchors. This design minimizes the severity of distribution shifts. Experimental results strongly demonstrate that this strategy can significantly accelerate convergence and reduce false hits on irrelevant concepts.
3、On the two highly representative benchmarks, UnlearnCanvas and I2P, POSDA has achieved the SOTA level in balancing the erasure accuracy (UA) and retention accuracy (IRA/CRA). Particularly in terms of retaining the general generation ability (FID/CLIP score), POSDA demonstrates remarkable stability.

Weaknesses
1、The author discusses the value of $\beta$ in the ablation experiment (Figure 8), but for different types of concepts (such as style vs. nudity), does there exist a universal recommended range of values?
2、The current experiments mainly focus on Stable Diffusion v1.4/1.5. With the widespread adoption of more advanced architectures such as SDXL and Flux, will POSDA be able to directly transfer to models based on the DiT (Diffusion Transformer) architecture? It is suggested that the authors conduct a brief discussion or provide preliminary verification on this matter.
3、The forgetting effect of POSDA is highly dependent on the accuracy of the classifier ϕ. If ϕ has biases or adversarial vulnerabilities, will it lead to forgetting failure or even generate new biases?

---

> ### Author Rebuttal · Authors · 2026-03-31
>
> We appreciate your thoughtful feedback and your recognition of our work. We have provided detailed clarifications for all the points you raised.
>
> **W1. Recommended beta range across concept types.**
> While Appendix B details task-specific optimal values, our empirical observations indicate a universal recommended tuning range of $\beta \in [10^{-7}, 10^{-5})$. The precise choice within this window fundamentally depends on the semantic complexity of the forget concept. Specifically, for localized or semantically bounded concepts (e.g., specific objects), a smaller $\beta$ is typically sufficient to preserve normal generation. Conversely, for highly generalized and deeply rooted concepts (e.g., NSFW content), a stronger alignment constraint is recommended to effectively suppress catastrophic forgetting.
>
> **W2. Transferability to DiT-based models.**
> We thank the reviewer for this forward-looking perspective. We confirm that POSDA is easily transferable to advanced models like SDXL, DiT, and Flux. Unlike many existing unlearning baselines that rely on manipulating specific internal modules (such as cross-attention matrices in U-Net), POSDA formulates the unlearning objective based on the output space of the network. Both our preference optimization and distribution alignment treat the diffusion backbone as a step-by-step mapping from noisy inputs to the denoising target, bypassing internal architectural bottlenecks.
>
> Specifically, for SDXL and standard DiT models, the mathematical parameterization of the transition probability $p_\theta$ and the noise predictor $\epsilon_\theta$ remains identical to SD v1.4/1.5. Therefore, POSDA is seamlessly applicable without algorithmic modification. For cutting-edge architectures like Flux that predict velocity field $v_t$ rather than noise $\epsilon$, our score-level alignment conceptually extends to velocity-level alignment. While establishing a strict mathematical bound for this extension requires further theoretical derivation under the ODE framework, the practical implementation of penalizing the divergence between the predicted vector fields remains directly applicable.
>
> **W3. Sensitivity to classifier performance.**
> We agree that POSDA’s unlearning efficacy is inherently tied to the accuracy of the reward classifier $\phi$. A severely biased or vulnerable classifier could theoretically misguide the optimization. However, from a practical standpoint, obtaining highly robust and accurate classifiers is highly feasible and cost-effective.
>
> First, for broad concepts, we can directly leverage state-of-the-art pre-trained Vision-Large Language Models. For instance, in our NSFW unlearning task, we directly utilized Qwen2.5-VL. These modern VLMs have undergone extensive safety alignment and RLHF, making them highly robust against common biases and adversarial inputs. Second, for specialized concepts, we can easily fine-tune lightweight classifiers based on strong pre-trained backbones with the training dataset to explicitly mitigate domain-specific biases.
> Furthermore, POSDA’s explicit modular design provides a structural advantage: if a vulnerability is discovered in the reward model, $\phi$ can be seamlessly updated or swapped for a stronger model without altering the core unlearning algorithm.

---

> > ### Author Rebuttal · Reviewer_5sed · 2026-04-07
> >
> > I appreciate the authors’ detailed rebuttal. The response partially addresses my concerns and improves several aspects of this paper, I will keep my score and don't have any other questions.

---

### Official Review · Reviewer_Vq3J · 2026-03-12

**Soundness:** 3
**Presentation:** 3
**Significance:** 3
**Originality:** 3
**Overall Recommendation:** 4
**Confidence:** 3

**Summary:**

This paper proposes **POSDA**, a unified framework for machine unlearning in text-to-image diffusion models. The method combines a preference-based optimization objective for concept removal with a score-level distribution alignment term intended to preserve behavior on retained concepts. Concretely, the paper formulates diffusion unlearning as an RL problem with a terminal reward that penalizes the forget concept while encouraging movement toward dynamically selected anchor concepts from the retain set. To reduce utility degradation, it further regularizes the unlearned model so that its score function remains close to that of the original model on retain prompts. Experiments on **UnlearnCanvas** and **I2P** suggest strong erasure performance together with improved utility retention, and the paper includes ablations on anchor selection and alignment strength. Overall, the paper tackles an important and timely problem. The combination of reward-driven unlearning and score-level preservation is interesting, and the empirical results are promising. That said, several technical and experimental aspects would benefit from a more careful treatment before the claims are fully convincing.

**Compliance With Llm Reviewing Policy:**

Affirmed.

**Final Justification:**

I will maintain my score

**Key Questions For Authors:**

1. Can the authors provide a clearer derivation and justification for Eq. (13), explicitly separating exact equalities from approximations?
2. For the dynamic anchor mechanism, how often are anchors recomputed during training and how many generated samples?
3. What is the computational overhead of POSDA relative to strong baselines, especially due to reward modeling and score-alignment updates?

**Limitations:**

yes

**Strengths And Weaknesses:**

### Strengths

1. The paper addresses two well-known issues in diffusion unlearning: the mismatch between surrogate training objectives and the actual unlearning goal, and the tendency of existing methods to damage retained capabilities. Recasting unlearning as preference optimization over denoising trajectories is an interesting idea, and the dynamic anchor mechanism is a sensible way to avoid unconstrained repulsion from the forget concept. The score-level alignment term is also well motivated as a stronger preservation mechanism than parameter- or image-level constraints.
2. The experimental results on UnlearnCanvas and I2P are strong overall. In particular, the method appears to achieve a favorable balance between erasure efficacy and retention, which is the main criterion that matters in this setting. The ablations on anchor size and alignment strength are useful and generally support the design choices.
3. The paper is generally well organized



### Weaknesses

1. For safety-related unlearning, it would be helpful to see robustness evaluations against adversarial prompts. The current I2P results show improved suppression of NSFW content under the benchmark prompts, but they do not tell us much about whether the erased concept can be recovered through prompt variation or jailbreak-style attacks.
2. The paper does not provide enough information about computational overhead, which is especially relevant given the use of external reward/classifier models and dynamic anchor selection.
3. The method appears to rely on concept-specific hyperparameter tuning, raising concerns about robustness and ease of deployment.

---

> ### Author Rebuttal · Authors · 2026-03-31
>
> We appreciate your thoughtful feedback and your recognition of our work. We have provided detailed clarifications for all the points you raised.
>
> **W1. Robustness against adversarial prompts and jailbreaks.**
> We appreciate this insightful suggestion. While we agree that adversarial defense is a critical frontier for unlearning, our current scope focuses on establishing the fundamental preference optimization framework. This aligns with recent SOTA baselines (e.g., MACE, EDiff, SPM, FMN), which primarily evaluate on standard benchmarks. We view adversarial robustness as a highly valuable extension for future work. Importantly, POSDA is naturally compatible with such extensions. Because the framework is reward-driven and modular, stronger reward models or adversarially generated prompts can be seamlessly incorporated into the preference optimization pipeline without altering the core algorithm. We will explicitly include a discussion on this limitation and outline this promising future direction in the revised manuscript.
>
> **W2 & Q3. Computational overhead analysis.** Thank you for your important suggestion regarding computational overhead. We have provided a comparison of the computational time for POSDA and the baselines in the Object Unlearning task. As shown in the table below, POSDA's computational overhead is comparable to that of the existing baselines, with no significant increase in overall time:
> | Method | ESD | EDiff | FMN | SAeUron | POSDA |
> | :--- | :--- | :--- | :--- | :--- | :--- |
> | Training Time (min) | 233.6 | 181.5 | 164.0 | 62.9 | 197.2 |
>
> Regarding the specific components raised: (i) The dynamic anchor selection and the external reward model do introduce additional computation. (ii) Computing the reference score for alignment requires a forward pass through the frozen original model. However, this is a standard regularization practice shared by most recent baselines (e.g., EDiff, FMN), POSDA incurs a comparable computational overhead in this aspect.
> The additional computational cost of POSDA is justified by its improved unlearning efficacy and better retention of non-target concepts. We will include this comprehensive computational analysis in the revised appendix.
>
> **W3. Concept-Specific Hyperparameter Tuning.** We address this deployment concern from two perspectives:
> First, concept-specific hyperparameter tuning is a widely adopted practice in the diffusion unlearning field (e.g., SAeUron, FMN). POSDA is consistent with the best practices in the field.
> Second, POSDA does not require exhaustive or unpredictable grid searches. As detailed in Appendix B, Table 3, the optimal hyperparameters fall within a bounded and predictable range. For example, the choice of the $\beta$ follows a clear semantic heuristic: localized concepts (e.g., specific objects) only require a smaller $\beta$, while broad, deeply rooted concepts (e.g., NSFW) necessitate a larger $\beta$. This predictable pattern ensures that POSDA remains highly robust and straightforward to deploy.
>
> **Q1. Clarification for Eq.13.**
> We thank the reviewer for the rigorous mathematical check.
> First, regarding Proposition 3.1, the true marginal score is exactly the expectation of the conditional score over the posterior via Tweedie's formula. Because computing this true expectation is intractable, we introduce our primary approximation by substituting it with the denoising network $\epsilon_\theta(x_t, c, t)$, which is trained via MSE to empirically estimate this posterior mean.
>
> Second, regarding the generative trajectory gradient, the implementation generates $x_0 = f_\theta(\xi, c_r)=\mu_\theta^{(1)}\circ \mu_\theta^{(2)}\circ \cdots \circ \mu_\theta^{(T)}(x_T,c_r,T)$. Thus, the exact gradient is computed via automatic differentiation through the full sampling trajectory. The single-step derivative $\frac{d\mu_\theta}{d\theta}$ used in the original Eq. (13) is a notational simplification for brevity. Accordingly, the exact formulation that reflects our actual implementation is
> $\nabla_\theta \mathcal{J}_{KL} (\theta) \approx \mathbb{E} \left[ \frac{\sqrt{\overline{\alpha}_t}}{\sqrt{1-\overline{\alpha}_t}} (\epsilon_{\theta_{old}}(x_t, c_r, t) - \epsilon_{\theta} (x_t, c_r, t)) \frac{\partial f_\theta}{\partial \theta} \right].$
>
> **Q2. Implementation details regarding anchor selection and data sampling.**
> First, regarding the recomputation frequency, the dynamic anchors are updated at every training iteration. As outlined in Algorithm 1, immediately after sampling a batch of forget concepts and generating their corresponding denoising trajectories, the anchor selection mechanism is executed dynamically to reflect the current policy state.
> Second, for each iteration, we sample both forget and retain concepts using a batch size of 8 across 16 inner rounds. Consequently, a total of 256 images is generated per iteration.

---

> > ### Author Rebuttal · Reviewer_Vq3J · 2026-04-01
> >
> > The rebuttal sufficiently addressed my comments. I don't have other questions.

---

> > > ### Author Response · Authors · 2026-04-04
> > >
> > > Thank you very much for your careful review and valuable comments on our work. We sincerely appreciate your time and constructive feedback.

---

### Official Review · Reviewer_rF2v · 2026-03-15

**Soundness:** 3
**Presentation:** 3
**Significance:** 2
**Originality:** 2
**Overall Recommendation:** 4
**Confidence:** 3

**Summary:**

The paper addresses the problem of unlearning concepts from text-to-image diffusion models. A new formulation is proposed with the unlearning objective being based directly on the posterior probability of generated images belonging to the unlearning concept. The authors then propose ‘anchoring’ the reward by dynamically selecting the retain concepts that are closest to the unlearn concept and then using a final reward designed to maximize alignment with anchor concepts while suppressing the unlearn concept. They further use a KL divergence regularization term to penalize deviation from the original distribution for the retain concepts. Experiments compare the unlearning and retaining performance of the proposed approach (POSDA) to many recent state-of-art baselines.

**Compliance With Llm Reviewing Policy:**

Affirmed.

**Final Justification:**

The clarifications provided in the rebuttal, especially the new tradeoff curve plot provided which clearly demonstrates the improved tradeoff attained by using POSDA convinced me to raise my score to 4 as my concerns have been mostly addressed.

**Key Questions For Authors:**

See weaknesses above.

My main concerns are the seemingly limited novelty (See weakness 2) and the lack of explicit comparison between the tradeoffs in the experiments (See weakness 4). If the authors address these (along with the other weaknesses) I would be open to raising my score.

**Limitations:**

yes.

**Strengths And Weaknesses:**

**Strengths**

1- The paper highlights an important shortcoming of existing unlearning approaches which is potentially excessive distribution shifts to achieve concept unlearning. The dynamic anchoring idea proposed in this paper is elegant, intuitive, and seemingly somewhat effective in practically addressing this issue.

2- The empirical evaluation is reasonably comprehensive, using different metrics (UA, IRA, CRA, FID) to evaluate unlearning and retain performance and a large number and variety of baselines are compared against.

3- The writing is concise and easy-to-follow and the main ideas are presented very clearly.

**Weaknesses**

1- Reliance on a pre-trained classifier is a limitation that many of the baselines don’t share. Also while not directly relevant to the main purpose of the paper, some additional information about the classifier used (at least in the appendix) would be helpful for reproducibility.

2- The novelty of the paper is limited to the dynamic anchoring idea which while elegant and useful, seems to be more of an incremental algorithmic improvement. The other parts of the paper like the KL divergence regularization are well-known approaches.

3- The empirical results are qualitatively unconvincing. For example, Figure 4 doesn’t convince me of the importance of anchors as the final results look similar regardless of whether an anchor was used or not. In general, there is a lack of qualitative comparison between POSDA and state of art methods.


4- More importantly regarding the empirical results, In order to support the claim that POSDA provides a better tradeoff between unlearning and retaining performance, I believe having plots comparing the tradeoff curves (obtained by varying $\beta$) for POSDA and the baseline methods is crucial. I believe many of the baseline methods have a similar hyperparameter controlling this tradeoff. For example EDiff seems to have comparable performance to POSDA in table 2. Without comparing the tradeoff curve for both methods directly it would be difficult to claim that POSDA has a better tradeoff.

---

> ### Author Rebuttal · Authors · 2026-03-31
>
> We appreciate your thoughtful feedback, which is highly valuable in improving the quality of our paper. We have provided detailed clarifications for all the points you raised.
>
> **W1. Reliance on classifiers and reproducibility.**
>
> First, regarding reproducibility, we clarify that the necessary details of the classifiers are provided in Appendix B ("Reward Model Configuration") of the original manuscript. Our current implementation utilizes fine-tuned ViT-Large for styles, ResNet-101 for objects, and pre-trained Qwen2.5-VL for NSFW content as reward models. We will add more implementation details in the revised manuscript.
>
> Second, regarding the reliance on a classifier, we view this as an architectural advantage rather than a limitation. Reliance on a reward model is inherent to preference-based optimization. By making this dependency explicit, POSDA directly aligns the optimization trajectory with the true erasure goal. This fundamentally resolves the critical "objective mismatch" problem that plagues surrogate-objective baselines (e.g., ESD, FMN). This modularity also allows our framework to seamlessly integrate more powerful reward models as they become available.
>
> **W2. Clarification of Core Novelty and Contributions.**
>
> We respectfully emphasize that our primary novelty is not limited to the dynamic anchor mechanism, but rather the fundamental paradigm shift that reformulates diffusion unlearning from heuristic surrogate guidance into a principled preference optimization (PO) problem over denoising trajectories. Existing methods rely on heuristic surrogate objectives, whereas our PO formulation allows for direct, quantifiable penalization of the target concept. The dynamic anchoring is a critical enabler of this formulation, constructing robust preference pairs in continuous generative spaces without degrading non-target generation quality.
>
> Furthermore, our use of KL divergence is fundamentally different from traditional approaches. Instead of conventional output-level or parameter-space regularizations, POSDA introduces a score-level distribution alignment. By applying KL divergence to match the score functions across intermediate diffusion timesteps, we provide a fine-grained, step-by-step constraint on the generative manifold. This score-level alignment captures the intrinsic dynamics of the diffusion process, preventing catastrophic forgetting far more effectively than standard regularizations. Therefore, the synergy of trajectory-based PO and score-level alignment constitutes a unified and conceptually novel framework.
>
> **W3. Clarification of qualitative results.**
>
> Importantly, the primary purpose of dynamic anchors is not to alter the final "safe" visual semantics, but to reshape the optimization gradient space and significantly accelerate unlearning convergence. Without anchors, the unlearning trajectory lacks explicit directional guidance in the latent space; with dynamic anchors, the forget samples are provided with structured, high-reward targets, dramatically accelerating convergence to safe regions. This is also consistent with the quantitative trend in Figure 5(b), where the unlearning efficacy (IRA) rises the fastest at $k=3$.
>
> Furthermore, we provide visual results for the "dog" unlearning task against SOTA baselines via this anonymous link: https://anonymous.4open.science/r/Anonym25338. As demonstrated, POSDA achieves a visibly superior unlearning-retention trade-off. We will include more detailed visual analyses in the revised manuscript.
>
> **W4. Tradeoff Curves and Baseline Comparisons.**
>
> We would like to clarify a technical detail regarding EDiff: it does not rely on a tunable scalar hyperparameter to control the unlearning-retention tradeoff, but instead uses an internal dynamic weighting mechanism. Therefore, generating a standard tradeoff curve by varying a hyperparameter is not directly applicable to EDiff.
>
> For all baselines evaluated in Table 2, we ensure a rigorous and fair comparison. We combine the original papers' recommended configurations with an extensive grid search to optimize their forgetting-retention balance, ensuring the reported results reflect their best performance. As shown in the existing quantitative results, POSDA demonstrates clear advantages across multiple tasks, particularly in terms of the average performance (Avg.).
> We will include comprehensive tradeoff curves comparing POSDA with strong, tunable state-of-the-art baselines (e.g., SPM) in the appendix of the revised manuscript.

---

> > ### Author Rebuttal · Reviewer_rF2v · 2026-04-02
> >
> > I thank the authors for their response. I apologize for having missed the classifier details in Appendix B.
> >
> > I have some follow-up questions.
> >
> > Regarding W2, I was under the impression that the standard diffusion model training objective is in fact KL divergence, and that minimizing the KL is equivalent to minimizing the score matching objective which essentially helps "match the score functions across intermediate diffusion time steps" as you put it. Is the regularization that you are using different? If so, how?
> >
> > Regarding W3, I now understand the purpose of Figure 4, however regarding your assertion that "Importantly, the primary purpose of dynamic anchors is not to alter the final "safe" visual semantics, but to reshape the optimization gradient space and significantly accelerate unlearning convergence.", the presentation of the paper does not seem to reflect this. In fact only a small part of the empirical results (the ablation in Figure 5) seems to be concerned with the acceleration of the unlearning convergence. I bring this up because now I am confused as to whether my main takeaway should be that POSDA accelerates convergence, or improves the retain-unlearn tradeoff, or both.  For the acceleration claim, I think more expansive ablations would be needed. For the tradeoff we come to W4.
> >
> > Regarding W4, I would like to see at least one tradeoff curve comparison. For example ESD I am sure has a parameter with which this tradeoff can be controlled. I say this because in optimizing forgetting-retention balance, what is the optimal solution? would this not be subjective depending on how much a given user cares about retention vs forgetting?

---

> > > ### Author Response · Authors · 2026-04-03
> > >
> > > We sincerely thank you for the highly constructive and insightful follow-up questions.
> > > We address each of your points in detail below:
> > >
> > > **1. Differences in Score Matching Regularization.**
> > >
> > > POSDA’s score-level distribution alignment differs fundamentally from standard diffusion training in its computational graph and gradient path.
> > >
> > > (i) **Computational Flow.**
> > > Standard diffusion training minimizes the KL divergence to the real data distribution from scratch, i.e. $\text{KL}(q(\mathbf{x}\_{t-1}|\mathbf{x}\_0) \| p\_\theta(\mathbf{x}\_{t-1}|\mathbf{x}\_t,c))$.
> > > Its computational flow applies forward noising to fixed real images: $\text{Real } x_0 \sim q\_{data} \xrightarrow{\text{add noise}} x\_t \xrightarrow{\text{single-step loss}} \text{Match } \epsilon$.
> > >
> > > In contrast, POSDA's alignment minimizes $\text{KL}(p_\theta(\cdot|c_r) \| p_{\theta_{\rm old}}(\cdot|c_r))$. It applies forward noising to images generated by the active policy: $\text{Noise } x_T \xrightarrow{\text{multi-step denoise } (f_\theta)} \text{Generated } x_0 \xrightarrow{\text{add noise}} x_t \xrightarrow{\text{alignment loss}} \text{Match } \epsilon_{\theta_{old}}$.
> > >
> > > (ii) **Gradient Path.**
> > > Because the starting point of the forward noising in POSDA is a self-generated image, the mathematical nature of the gradient updates changes entirely.
> > > In standard diffusion training, $x_0$ denotes the real image (a fixed constant), the gradient only updates the network parameters directly at timestep t via a simple L2 regression.
> > > In POSDA, $x_0$ is the generated image, parameterized by the current model $\theta$ through the reverse sampling process ($x_0 = f_\theta(x_T, c_r)$). Rigorous KL minimization requires the gradient to backpropagate through the entire multi-step sampling trajectory (corrected Eq. 13):
> > > $\nabla_\theta\mathcal{J}\_{KL}(\theta) \propto \mathbb{E}\left[\frac{\sqrt{\bar\alpha\_t}}{\sqrt{1-\bar\alpha\_t}} (\epsilon\_{\theta\_{\rm old}} - \epsilon\_\theta) \nabla\_\theta f\_\theta \right]$.
> > >
> > > This makes POSDA’s alignment **a trajectory-aware gradient update**. Instead of learning a score function from scratch via single-step noise regression, it physically **reshapes the generative manifold by pulling the full sampling trajectory** based on the score discrepancy, effectively anchoring the original generative dynamics.
> > >
> > > **2. The Main Contribution and Role of the Reward.**
> > >
> > > To clarify, our main takeaway is the methodological shift to a unified framework featuring two parallel pillars, i.e., Preference Optimization for targeted unlearning and Score-Level Distribution Alignment for retention, which ultimately results in the superior tradeoff. Within this framework, PO's primary contribution is abandoning heuristic surrogate proxies in favor of an explicit reward that directly quantifies the unlearning objective.
> > >
> > > The dynamic anchor is not the overarching framework itself, but rather a crucial component of the reward design, motivated both heuristically and empirically.
> > > (i) Heuristically, if we only penalize the forget concept, the unbounded optimization may destroy the surrounding data manifold. The anchors are introduced specifically to provide a safe target region within the reward function.
> > > (ii) Empirically, our preliminary experiments revealed that without anchors, images generated from the forget prompt eventually converge to a few fixed retain concepts. However, the intermediate generations are highly chaotic and unstructured.
> > > Therefore, we introduce dynamic anchors to provide a direct shortcut to these safe concepts, eliminating the chaotic exploration phase.
> > >
> > > **3. Trade-off Curve Comparison and Optimality.**
> > >
> > > We agree that the optimal forgetting-retention balance is inherently subjective and depends on the specific application [1]. However, in experiments, we follow [2-3] and select the best configurations based on Pareto optimality: a solution is optimal if one cannot improve forgetting without severely degrading retention.
> > > Consequently, to quantitatively reflect this holistic tradeoff in our main results, we reported the Avg. of UA, IRA, and CRA (in Tab.1).
> > >
> > > We follow your excellent suggestion and plot the tradeoff curves (UA vs. mean of IRA and CRA) for the style unlearning task.
> > > We compared POSDA against ESD and SPM, by varying their respective unlearning-retention control hyperparameters.
> > > The resulting tradeoff curves have been added to https://anonymous.4open.science/r/Anonym25338/trade_off_curve.png.
> > > We highlight the optimal operating point for each method with a black outline, representing the configuration that maximizes both objectives before severe degradation occurs.
> > > Visually, POSDA's tradeoff curve and optimal point lie significantly closer to the ideal top-right corner, establishing strict Pareto dominance over both baselines.
> > >
> > > [1] Erasing Concepts from Diffusion Models
> > >
> > > [2] Erasing Undesirable Influence in Diffusion Models
> > >
> > > [3] Direct Unlearning Optimization for Robust and Safe Text-to-Image Models

---

### Decision · Program_Chairs · 2026-04-30

**Decision:**

Accept (regular)

**Comment:**

This paper proposes a novel approach, namely Preference-calibrated Optimization with Score-level Distribution Alignment (POSDA), to address the problem of machine unlearning. The contribution is solid, and following the rebuttal, the reviewers have reached a consensus with an overall positive assessment.

Based on these evaluations, I recommend acceptance of the paper. I encourage the authors to carefully revise the manuscript in line with the reviewers' feedback, particularly by improving the clarity of the methodology and providing more detailed derivations.